# Data in Context: How Digital Transformation Can Support Human Reasoning in Cyber-Physical Production Systems



**Romy Müller** [1,*]**, Franziska Kessler** [2]**, David W. Humphrey** [3] **and Julian Rahm** [4]

1 Faculty of Psychology, Chair of Engineering Psychology and Applied Cognitive Research, Technische Universität Dresden, 01069 Dresden, Germany
2 Faculty of Psychology, Chair of Learning and Instruction, Technische Universität Dresden, 01069 Dresden, Germany; franziska.kessler@tu-dresden.de
3 ARC Advisory Group, 80999 Munich, Germany; dhumphrey@arcweb.com
4 Faculty of Electrical and Computer Engineering, Chair of Process Control Systems & Process Systems Engineering Group, Technische Universität Dresden, 01069 Dresden, Germany; julian.rahm@tu-dresden.de
* Correspondence: romy.mueller@tu-dresden.de; Tel.: +49-351-463-35330

**Abstract:** In traditional production plants, current technologies do not provide sufficient context to support information integration and interpretation. Digital transformation technologies have the potential to support contextualization, but it is unclear how this can be achieved. The present article presents a selection of the psychological literature in four areas relevant to contextualization: information sampling, information integration, categorization, and causal reasoning. Characteristic biases and limitations of human information processing are discussed. Based on this literature, we derive functional requirements for digital transformation technologies, focusing on the cognitive activities they should support. We then present a selection of technologies that have the potential to foster contextualization. These technologies enable the modelling of system relations, the integration of data from different sources, and the connection of the present situation with historical data. We illustrate how these technologies can support contextual reasoning, and highlight challenges that should be addressed when designing human–machine cooperation in cyber-physical production systems.

**Keywords:** operator assistance systems; cyber-physical production systems; contextualization; cognitive psychology; digital transformation; information modelling





## 1. Introduction

### 1.1. The Challenges of Contextualization in Industrial Production Plants

In the process industries and discrete processing industries, operators' process monitoring and process control activities can be characterized as problem solving [1–3]. First, much of the available information is irrelevant. For instance, in the process industries, operators often receive about 50 alarms per hour [4] and data are typically presented as a large number of individual values. As plant behavior is subject to the causal constraints resulting from natural laws (e.g., chemical, thermodynamic), these values are interdependent and not all combinations can occur [5]. Thus, it is possible to derive sufficient information from only a few indicators. However, this requires operators to know which ones to check in what situations, because the importance of data depends on the current production context.

A second, complementary problem is that much of the relevant information is unavailable. For instance, in many discrete processing plants, information from previous production steps is not transmitted to operators of subsequent steps. As production steps are highly interdependent, such information would provide the necessary context to interpret current data and predict future developments. To understand what we mean by context, consider an operator supervising a machine that packages chocolate bars. The machine is subject to frequent faults and stoppages, which result from a complex interplay of different factors [2]. These factors are not restricted to the operator's packaging machine

but also concern the broader production context: parameter settings and events in the molding unit, the resulting characteristics of chocolate bars, characteristics of the packaging materials, and environmental conditions. Therefore, operators should know about this context and its causal relationship with their own machine. Usually, such knowledge and data are not available. Hence, appropriately contextualizing the available data and observations is a major challenge.

The challenges of contextualization can be specified in terms of the following context-dependent cognitive activities: (1) *Sampling the available information*: What subset of the huge amount of data is relevant? (2) *Integrating different information elements*: How should data from different sources be combined to form a coherent picture? (3) *Categorizing objects and events*: How should situations be interpreted and compared? (4) *Reasoning about causes*: What are the causes and consequences of a currently observed data pattern?

These cognitive activities remain challenging in today's plants, as contextualization is barely supported by contemporary technologies. Although the Human Factors literature has suggested interface concepts to address some of the challenges, e.g., [6–9], these concepts cannot solve the contextualization problem in real plants. One reason for this is that current technologies do not sufficiently support their implementation [10]. This is expected to change in cyber-physical production systems (CPPS).

## 1.2. Digital Transformation and Human–Machine Cooperation in CPPS

A central aspect of Industry 4.0 and CPPS is the merging of the physical world with the virtual world: CPPS are characterized by intensive connections between collaborating computational entities and physical objects, relying on large quantities of data and using a variety of services to access and process these data [11]. In this context, the exchange of information plays a significant role [11,12]. For instance, planning data from different disciplines are made available throughout the entire plant lifecycle, a close communication between different machines or devices is established, and current sensor data are continuously analyzed and applied for control purposes. This technological innovation of CPPS rests on a wide variety of concepts (e.g., formal semantics, information modelling standards) and specific technologies (e.g., ontologies and Linked Data, OPC UA). For instance, semantic networks of various planning data [13] can be used to model system relations, and semantic communication standards for the processing of live data [14] can be used to integrate data from different sources. As a result, the boundaries within the classic automation pyramid are increasingly dissolved, and information systems are loosely coupled in order to respond more flexibly to new requirements.

Such digital transformation technologies for information modelling, processing, exchange, and integration in CPPS have the potential to make relevant context information available wherever it is needed in the system. On the one hand, in this context, information can enhance system control and automatic failure detection within the system [15]. On the other hand, it can be provided to human operators to support their context-dependent cognitive activities, and thereby enhance human reasoning in unprecedented ways.

## 1.3. Aims of the Present Work

In the present article, we ask how contextualization can improve human–machine cooperation: How can digital transformation technologies support the context-dependent selection and integration of data from different sources, and thereby support categorization and causal reasoning? We focus on supervisory control tasks that require operators to monitor highly automated processes and intervene if necessary [16]. To specify how context-dependent monitoring activities can be supported, we first provide a selection of literature from different psychological areas. This literature overview is organized according to four challenges that operators must face to contextualize data (i.e., information sampling, integration, categorization, and causal reasoning). Based on this literature, we extract requirements for technical support and provide an overview of digital transformation technologies for information modelling, distribution, and integration. We ask how these

technologies could address the issues identified in the psychological literature section and discuss the challenges of their application.

On a more abstract level, we discuss how technology can address the problems described in the psychological literature, and what cognitive requirements must be considered to do this successfully. This focus differs from a large body of the Human Factors literature, which argues that technological innovation creates new problems for humans. For instance, the psychological literature on automation effects has engaged in the latter reasoning for decades, e.g., [17–19], and recently this work has been extended to novel digital transformation technologies, e.g., [20–23]. Although this perspective has revealed many important insights, the present article takes the opposite approach, considering technology as an enabler rather than a problem source, and asking how it can be used in a way that benefits human cognitive activities.

The potential of technology as an enabler of human–machine cooperation in CPPS has been discussed in previous work on the concept of "Operator 4.0", e.g., [24–26]. The present article differs from this work in three ways. First, we focus on a specific aspect of human cognition (i.e., interpreting data in context) and base our requirements for technologies on empirical findings from the corresponding psychological literature, whereas previous work has provided a general overview of operator assistance with issues as diverse as collaboration, health, and physical strength. Second, we focus on information contents, whereas previous work has often emphasized the presentation medium (e.g., virtual and augmented reality). Based on the first two points, we focus on technologies for information modelling and the integration of process data, whereas previous work has focused on intelligent spaces that measure operator activities, states, and tacit knowledge (e.g., smart sensors, wearable devices). Taken together, the scope of the present article is narrower and more specific than that of previous related work.

At the same time, it is impossible to provide a comprehensive description of all relevant research in just one article. This concerns both the psychology section and the technology section of the article. In the psychology section, the selection of the four research areas, the 45 component issues in these areas, and the individual studies used to illustrate these issues only represents a fraction of the relevant literature, because the field of cognitive psychology is immensely large. A thorough consideration of even one of the research areas could easily fill a textbook, and each of the component issues raised in these areas could be the topic of its own systematic review. In the technology section, we only present relevant areas of technological development, instead of specifying how to apply individual technologies. This will not allow the reader to extract ready-made solutions, as the devil is in the details. However, going into the depth needed to specify solutions would only allow us to cover one psychological issue or one technology.

Instead, we provide a broad overview of issues to consider when using digital transformation technologies to design human–machine interaction. This choice was made because cooperation between human-centered and technology-centered disciplines is more important than ever in the context of CPPS, where more and more data are available to operators, and technologies are increasingly capable of taking over cognitive tasks. However, engineers cannot be expected to read dozens of psychological research papers, and vice versa, as a precondition for such cooperation. Therefore, our aim is to build bridges between disciplines, and the broad approach of this article should serve as a starting point for an interdisciplinary discussion about the cognitive potential of digital transformation.

## 2. Cognitive Challenges of Reasoning in Context

Although "contextualization" is not a psychological construct in itself, different areas of psychology are relevant with regard to the question of how people put information in context when trying to make sense of the world. In the following section, we specify what this means by presenting examples of empirical studies from four areas of psychology that investigate how people sample and use available information, how they integrate

different information elements, how they categorize objects and events, and how they reason about causes.

*2.1. Sampling the Available Information*

How do people sample from the large pool of available information and how do they use these samples to draw conclusions about the world? For instance, will operators only check whether chocolate bars deviate from their optimal state in case of a fault, or will they also consider the base rates of such deviations? Which types and sources of information will they consider in what situations? The following sections provide evidence that people do not always generate unbiased, representative information samples, and often they are unaware of their sampling biases. Information search is highly selective and people ignore particular types of information. Moreover, search is modulated by characteristics of the tasks and information sources.

2.1.1. Information Samples Are Biased

People are quite good at drawing inferences from a given sample, but bad at judging whether their samples are representative [27]. The notion of *sampling biases* differs from reasoning biases in that it stresses that bias already is present in the information sources. Accordingly, Fiedler and Kutzner [28] (p. 380) conclude that, "in order to understand the cognitive processes within the decision maker one first of all has to analyze the structure and the constraints of the information input with which the decision maker is fed". Biased information sampling is partly responsible for commonly reported reasoning biases such as the *availability heuristic*—the fact that people judge events as more common when they come to mind easily [29]. That is, when particular events are overrepresented in the available information, they are considered to be more likely than they actually are. Although this is correct based on the available sample, it still leads to a distorted representation of the actual state of the world.

Sampling biases often result from *conditional sampling* [28]. For instance, operators might only check whether chocolate bars have geometrical distortions after a machine stoppage. However, this strategy neglects the base rate of geometrical distortions, and thus inflates their perceived impact. Sampling biases can also result from the fact that people *repeat choices that initially led to good outcomes* [30]. For instance, if operators have experienced that monitoring machine cooling allowed them to detect a critical event early, they are more likely to sample this parameter again than if they have initially experienced that it was not helpful. In the latter case, they might never find out that machine cooling actually is important. Sampling biases can be *counteracted* by changing the presentation of information, for instance, by making it transparent how the sample was selected [31] or by asking people to find arguments against an anchor or standard, which mitigates the selective accessibility of consistent information [32].

2.1.2. Selecting and Ignoring Particular Types of Information

People selectively use particular types of information while systematically ignoring others. They often rely on *salient cues* or information elements even when they are invalid [33], and focus on the extremeness of evidence instead of on its weight or validity [34]. For instance, operators might overinterpret high temperature deviations instead of asking how important temperature is in the current context.

When testing hypotheses, people are prone to *confirmation bias* [35]: They selectively look for information that supports their hypothesis, especially when the amount of available information is high [36]. However, some phenomena that look like confirmation bias actually result from a *positive test strategy*, meaning that people preferably test cases that have the property of interest [37]. For instance, when trying to find out whether low temperatures cause chocolate bars to break, operators are more likely to sample situations with low temperature or broken bars than situations with normal temperature or bars that remained intact. This leads to unequal sample sizes for different parameters and

outcomes, and thus can result in illusory evidence for the hypothesis [28]. Not only is information search biased towards the preferred hypothesis but also the way people react to hypothesis-consistent or inconsistent information, and novices, especially tend to *re-interpret information* to fit their hypotheses [38].

In contrast, people often ignore information that contradicts their hypotheses or that they were not actively searching for, and unexpected information can easily be overlooked. Such *inattentional blindness* [39] also occurs when people receive automated decision support in process control environments: they fail to cross-check information, especially when the degree of automation is high [40]. When evaluating situations, people sometimes do not take information about *contextual constraints* and possible negative consequences into account [41,42]. This even is the case for very simple, well-known information. Finally, people tend to focus on what they know instead of what they do not know, and this *neglect of unknown or missing information* can lead to overconfidence [43].

### 2.1.3. Tasks and Information Sources Affect Information Search

Higher *task complexity* leads people to use different information sources more often, both when operationalized as the dependence of jobs on environmental factors [44] and as the degree to which task outcomes, process, and information requirements are uncertain [45]. In the latter study, task complexity also affected which types of information sources people used. For simple tasks, they mostly relied on information about the structure, properties, and requirements of the problem (e.g., problem location). For more complex tasks, they increasingly used information about the domain (e.g., concepts or physical laws) and the methods of how to formulate and treat a problem (e.g., pros and cons of different strategies).

A second task characteristic that shapes information search and usage is the amount of available information. *Information overload* can negatively affect information processing and decision making (for a review, see [46]). For many people, the relation between information quantity and information usage follows an inverted U-shape [47]. When too much information is present, this leads people to ignore or misuse the available information [48]. Misuse can be reflected in increased error tolerance, source misattribution, incompletely representing the message, or abstracting its meaning. In consequence, information overload increases processing time [49] and reduces decision quality [47], especially under time pressure [50]. It also makes people less confident in their decisions [51], and can even lead them to refrain from making a decision altogether [52].

A third influence on information search is the *accessibility* of information sources [44]. Although people rate the importance of information sources according to their quality, the actual frequency of use largely depends on accessibility [53]. However, in order to generate testable theories and design operator support, it is important to define what accessibility means [54]. For instance, it can mean that information has the right format, the right level of detail, saves time, or that lots of information is available in one place. A particularly important factor associated with accessibility is the *familiarity* of information sources, and people most often select the sources they are familiar with.

### 2.1.4. Summary

Information samples are often biased, and people usually do not take these sampling biases into account when making inferences. Biased samples emerge when people only sample information in particular situations, are highly selective in the types and sources of information they use, and miss or neglect important information. This selectivity in sampling further increases when abundant information is present or information is hard to access. However, people aim to acquire more information when tasks are more complex, and technologies should support them in drawing appropriate samples from different sources.

*2.2. Integrating Different Information Elements*

How do humans combine information from different sources (e.g., previous production step, product characteristics, environment) to generate an overall model of the situation? This issue has mainly been studied in the context of decision making, where different information elements (i.e., cues) provide evidence for one of several options. In this context, people's information integration strategies differ with regard to whether the integration proceeds in a systematic manner and how much of the available information is used. Moreover, information integration is influenced by task and information characteristics.

2.2.1. Strategies of Information Integration

Cues can be integrated in a *formal or informal* manner. Formal integration means that cues are combined algorithmically, according to clearly specified rules. Informal integration means that they are combined in a subjective, impressionistic way. People usually believe they are better integrators than algorithms, because their experience allows them to weight and combine cues more appropriately [55]. However, a meta-analysis of 136 studies revealed that, on average, algorithms are 10% more accurate [56]. The effect size varies with factors such as the type of prediction, the setting of data collection, the type of formula, and the amount of information [57]. Under two opposing conditions, people perform significantly worse than algorithms [58]: in low-validity environments that are noisy or overly complex, so that humans cannot detect weak regularities (e.g., large samples of process data), and in high-validity environments that are almost entirely predictable, and thus can easily be analyzed by algorithms, while humans experience occasional attention lapses (e.g., automatic control of product flow). However, it should be noted that algorithms can only generate accurate outcomes when fed with appropriate data. This is an important constraint in production processes where some process states cannot be measured but only perceived by human operators [2].

Another distinction concerns *rule-* versus *exemplar-based* integration. The former means that cues are integrated based on cue–criterion relations (e.g., how strongly temperature and machine speed predict broken chocolate), and the latter means that the criterion values of exemplars with similar cue patterns are retrieved from memory (e.g., how often chocolate broke in the past when temperature and machine speed were similar). When people have sufficient knowledge about the cues, they prefer rule-based integration, whereas, when such knowledge is difficult to gain, they rely on exemplar-based integration [59]. Exemplar-based integration is also used as a backup strategy when cue abstraction (i.e., inferring the predictive power of cues) is difficult [60]. In consequence, strategy selection depends on the type of cue [61]: when cue abstraction is easy, with present/absent cues (e.g., geometrical distortions of chocolate bars or not), people prefer rule-based strategies, whereas, when cue abstraction is difficult, with alternative cues (e.g., geometrical distortions vs. soiled conveyor belt), they use exemplar-based strategies.

When making decisions based on a set of cues, people usually do not combine all cues weighted by their importance. Instead, they use simpler *heuristics*. For instance, they eliminate options with low values on the most important cue, select the option that performs best on most cues, or select the option that performs best on the most important cue [62]. Such heuristics can be ecologically rational, and thus lead to good outcomes in particular decision contexts [63–65].

2.2.2. Task and Information Characteristics Affect Integration

The selection of information integration strategies depends on task characteristics. With higher *task complexity*, people are more likely to rely on heuristics [66]. However, more information does not necessarily make decisions more difficult, and can even be processed faster than less information when it leads to higher *coherence* (i.e., all information elements fit together) and, thus, information can be integrated holistically [67]. The selection of integration strategies also depends on the *validity of easily accessible information* [68]. When easily accessible information has high predictive validity, people stop searching and choose

the option that performs best on the respective cue (i.e., *take-the-best heuristic*). Instead, when easily accessible information has low validity, they integrate all available information. Similarly, information integration depends on *presentation format*: when all relevant information is presented at once, people integrate it in a quick, holistic manner, while, when it is presented sequentially, they use simpler strategies [69]. However, even when all information is presented at once, strategy selection depends on the need for information search imposed by a given presentation format [70]: with higher search demands, people are less likely to integrate information holistically. Finally, *time pressure* drives people to use simple heuristics that ignore parts of the available information [62,71].

### 2.2.3. Summary

Information can be integrated more or less systematically. Algorithms usually outperform humans, but this depends on the complexity of the environment and whether all the relevant information is measurable. People can integrate information by relying on rules or exemplars, and the latter is used as a backup strategy when rule-based integration is not feasible. They often rely on heuristics, which can lead to good outcomes under certain conditions. In contrast, people can integrate information holistically when data are coherent, its presentation minimizes search demands, and sufficient time is available. Thus, technologies should either take over the integration, provide data in ways that facilitate integration, or make past exemplars (i.e., cases) available when situations are too complex to integrate all relevant data.

### 2.3. Categorizing Objects and Events

Based on the information that people sample and integrate, they can categorize objects and events. For instance, operators might categorize a conveyor belt soiled with chocolate as either a temperature problem or a problem that reduces machine efficiency. A large body of the literature describes how people form categories and concepts (for a recent discussion, see [72]). It tackles issues such as whether people rely on prototypes or exemplars, or how specific they are in their categorizations. As reviewing this literature is beyond the scope of this article, we only focus on two complementary functions of categorization that are relevant to dealing with context information: differentiation and generalization. Differentiation enables people to acquire new knowledge while also retaining previous knowledge, and decide which is needed in what contexts. Generalization enables them to detect similarities between contexts, allowing them to go beyond the information that is given. As the implications of these complementary functions for the process industries have been discussed elsewhere [22], this article only focuses on two aspects: the flexibility of categories and the role of similarity.

### 2.3.1. Differentiation: Context-Dependent, Flexible Categorization

To deal with a wide variety of situations, people need *flexible category representations* [73,74]. That is, knowledge must be interconnected, accessible from different perspectives, and stored in multiple cases or analogies. People must be able to decompose and reassemble their knowledge representations to deal with different situations, and consider several alternative interpretations of a situation. While abstract schemas that go beyond the specifics of a particular situation are necessary, the construction of these schemas relies on concrete cases [73,75].

Flexibility in categorization is important for short and longer timescales. On short time scales, people need to frequently update their knowledge about *environmental contingencies*. First, this requires them to detect situation-specific changes in the regularity of events [76,77]. For instance, when particular settings of parameter values lead to broken chocolate during the production of milk chocolate but not dark chocolate, operators must detect these changed contingencies and differentiate the two situations accordingly. Second, they must detect changes in the relations between interacting parameters such as temperature, moisture, and machine speed. To do so, they need to predict metric outcomes

and adapt these predictions to the current context. For instance, they need to realize when in some process state, temperature loses its predictive power while machine speed becomes more important. People are surprisingly good at this and can flexibly use different parameters according to their current influence [78].

On longer timescales, it can be necessary to change the concepts people use to categorize situations. Such *conceptual change* is difficult and prior knowledge can create conflict in three ways [79]: People may have false beliefs, rely on flawed mental models, or use wrong ontological categories. The latter is particularly hard as it requires people to fundamentally change their understanding (e.g., heat transfer is an emergent process and not a relocation of hot molecules). Learning information that contradicts prior knowledge is comparably easy when only cue values change but hard when the relevance of cues changes, so that previously irrelevant information has to be considered [80].

### 2.3.2. Generalization: The Role of Similarity

One function of categorization is that the acquired knowledge can be used later in similar situations, allowing people to go beyond the given information and infer unseen attributes [81]. However, the notion of similarity is *context-dependent* [82]. For instance, grey is judged as more similar to white than black for hair, while the opposite is true for clouds [83]. Moreover, similarity often cannot be assessed from obvious features of objects or events but depends on the *focus of attention*. For instance, are children similar to jewelry? Certainly not in terms of appearance or behavior, but perhaps when asking what things should be saved from a burning house [84]. In this case, attention might be directed to features such as "valuable", "irreplaceable" and "portable".

When judging the similarity of situations, attention is often captured by *surface features* while neglecting *structural features* [85]: people consider situations as similar when they share the same elements (e.g., both involve caramel chocolate) but fail to notice when situations share the same relations between elements (e.g., in both situations the contact between chocolate bars and the conveyor belt is impaired, but one time due to geometrical distortions and one time due to a clogged vacuum filter). Accordingly, categorization based on relations is more demanding [86] and domain expertise goes along with an increased reliance on relations instead of features [87].

However, it needs to be noted that categorization also depends on *features beyond similarity*. For instance, it is affected by people's theories about the world [88]: people are likely to judge a person who jumps into a pool fully clothed as drunk. Although there probably is no a priori association between the category and the specific behavior, it is in line with people's theories about the effects of being drunk. Other features that affect categorization are people's goals or the distribution of information within a category (for a review, see [82]).

### 2.3.3. Summary

People categorize objects and events to differentiate between situations and generalize across situations. Differentiation requires people to use flexible category representations and update their knowledge about environmental contingencies. Conceptual change may be needed, which is difficult, especially when previously irrelevant information becomes relevant. To generalize across situations, people must determine what similarity means depending on the current context and focus of attention. People are often misled by surface similarity and ignore the relationships between objects. Therefore, technologies should support operators in detecting and evaluating both context-specific changes, as well as structural similarities between situations.

### 2.4. Reasoning about Causes

Knowledge of the causal impacts of events is essential to explain, control, and predict the behavior of a plant. It can simplify decisions as it enables people to reduce the abundance of available information to a manageable set [89]. For instance, when people

know which parameters could be responsible for a fault, this allows them to focus on these relevant parameters and ignore others. To infer that events are causally related, people use different cues, such as covariation, temporal relations, and prior knowledge. In the following sections, we discuss how people use different cues to infer causality and how they deal with complexity while making such causal inferences.

### 2.4.1. Covariation

A first indication of causality is covariation [90], meaning that one event reliably follows another (i.e., the probability of the outcome is higher when the potential cause is present than when it is absent). However, covariation-based causality ratings are biased, for instance, by the *magnitude of probabilities*. First, people's causality ratings increase with the base rate or total probability of the outcome [91]. Second, people even infer causality when the outcome in the presence of a potential cause is as likely as the same outcome in the absence of that cause (illusory correlations) [90]. Moreover, people are prone to *inattentional blindness for negative relations* [92]. Taken together, these findings imply that causality ratings can be distorted by salient cues.

The presence of alternative potential causes can either decrease or increase causality ratings [93]. A situation in which it decreases them is *overshadowing*: if event A is associated with an outcome but always appears together with event B, the causal influence of A is rated lower [94,95]. A situation in which an alternative cause increases causality ratings is *super-learning*: if event A together with event B leads to an outcome, event C leads to the same outcome, but the combination of B and C do not lead to that outcome, the causal influence of A is rated higher [96]. That is, event A is assumed to be extremely powerful, as it can counteract the negative influence of B. The perceived influence of alternative causes also depends on the *type of task*: people are sensitive to the strength of alternatives in diagnostic reasoning from effect to cause, but not in predictive reasoning from cause to effect [97].

### 2.4.2. Temporal Relationships

Solely relying on covariation to infer causality is problematic [98]. One reason for this is that covariation does not provide information on the direction of the effect. Therefore, it is important to consider time as an indicator of causality. Causality ratings are shaped by two temporal factors: order and contiguity. As causes precede their effects, the *temporal order* of events is a powerful indicator of causality and can even dominate covariation [99]. However, temporal order can also be misleading (e.g., lightning does not cause thunder).

The second temporal factor is *temporal contiguity*, or the close succession of two events. Contiguity increases causality ratings, while long delays decrease them [100]. Moreover, causality ratings decrease with a high *temporal variability* of the delay between cause and effect [101]. The influence of delays is particularly important in the process industries, where delays and dead times are common and it can take hours or even days for an action to have an effect [102]. In the discrete processing industry, delays also play a role. For instance, in chocolate production, the effects of changes to the molding unit are only observable in the chocolate bars after about one hour. However, delays do not impair causality ratings when people *expect longer delays*, as they are aware of the underlying causal mechanisms and thus can understand why it takes time to generate an effect [103]. Such temporal assumptions determine how people choose statistical indicators for causality, select potential causes, and aggregate events [104]. This emphasizes the important role of prior knowledge in causal reasoning.

### 2.4.3. Prior Knowledge

Causal reasoning is guided by prior assumptions, expectations, and knowledge [105–107]. People have assumptions about the *causal roles of events*, which affects how they perceive contingencies [108]. Based on these mental models, they can even estimate the relationship between events they have never experienced together [109,110]. This is important, as

people typically only observe fragments of causal networks. To integrate these fragments, they have to *select appropriate integration rules*. For quantities that depend on amount (e.g., temperature of chocolate resulting from different sources) people add the impact of different causal influences, whereas for quantities that depend on proportions (e.g., taste of chocolate resulting from the combination of ingredients), they average causal influences [110]. Moreover, the selection of integration rules depends on domain knowledge and context factors such as data presentation, task type, and transfer from other tasks.

When reasoning about a specific domain, thinking in terms of cause and effect is too abstract [111]. For instance, operators need to conceptualize the relationships between process elements as feeding, opening, sucking, allowing, or speeding. It cannot be taken for granted that people have valid causal models, but this is a matter of *experience*. While domain experts are able to reason about things in terms of causal phenomena (e.g., negative feedback), novices are more likely to classify them by surface features [112].

### 2.4.4. Dealing with Complexity

Industrial plants are complex systems characterized by dynamic and partly intransparent interactions between many variables. In such systems, there are several limitations to people's processing of causal information. For instance, people often engage in *linear reasoning* and do not take complex system features such as emergence or decentralization into account [113]. While experts and novices do not differ considerably in their understanding of system structures, novices have serious difficulties in *understanding causal behaviors and functions* [114]. Although people typically believe that they know about the mechanisms of complex systems, this knowledge is often imprecise, incoherent, and shallow [115]. This *illusion of understanding* can be traced back to the fact that people think about systems in too abstract terms [116].

### 2.4.5. Summary

When reasoning about causes, people use different cues such as covariation, temporal relations, and prior knowledge. These cues are noisy indicators of causality and none of them are completely reliable, but, in combination, they can provide compelling evidence about the causal relations underlying the observed data [117]. However, people often think in concepts that are too abstract, and find it difficult to understand complex system concepts. Therefore, technologies should make the relationships between objects and events understandable, support operators in reasoning about causes, and help select appropriate integration rules despite disturbing factors such as time delays and interactions.

## 3. Requirements: What Should Technologies Do to Support Humans?

For each cognitive challenge identified in the previous section, we derived possible ways of addressing it. These *general requirements for operator support strategies* do not yet specify any technology. Altogether, we extracted 108 support strategies. In Table 1, we present two examples for each psychological area (for a full list, see Appendix A, Tables A1–A4). For instance, considering that people are prone to conditional sampling, it should be made explicit when events are overrepresented in samples. This could be done by informing operators that a particular geometry distortion of chocolate has only been observed when a machine fault occurred, which might overestimate its impact on the production process.

**Table 1.** Examples of deriving requirements for support strategies from the cognitive factors presented in Section 2.

| Cognitive Factors | General Requirements for Operator Support Strategies | Examples |
|---|---|---|
| Sampling the available information | | |
| Conditional sampling | Make it explicit when events may be overrepresented in samples | "Chocolate bars have mostly been checked for hollow bottoms when faults have occurred, which may overestimate the impact of this distortion" |

**Table 1.** *Cont.*

| Cognitive Factors | General Requirements for Operator Support Strategies | Examples |
|---|---|---|
| Availability heuristic | Make information available in an unbiased manner | Provide different data sources (e.g., temperature in end cooler, buffer, and packaging machine) and data types (e.g., temperature, soiling, and motor currents) |
| Integrating different information elements | | |
| Rule- versus exemplar-based | Support cue abstraction (i.e., make predictive power of cues explicit) | "Hollow bottoms are the strongest predictor of skewed chocolate bars" |
| Relying on heuristics | Support operators in assessing the context-dependent suitability of heuristics | "To determine whether there was a problem with a packing claw, it is sufficient to check whether every eighth bar was affected" |
| Categorizing objects and events | | |
| Environmental contingencies | Highlight changes in relations between interacting parameters | "High temperature is less problematic now, because you have reduced machine speed" |
| Conceptual change | Provide factual information to help operators detect and correct false beliefs | "The turning wheel is not responsible for squished chocolate bars" |
| Reasoning about causes | | |
| Overshadowing and super-learning | Make interactions between causes explicit (e.g., additive, enhancing, suppressing) | "The effects of low temperatures in the molding unit are cancelled out when chocolate bars remain in the buffer for a long time" |
| Temporal variability | Provide information about factors affecting the variability of delays | "The delay of molding problems affecting the packaging machine depends on the time that chocolate bars remain in the buffer" |

These general requirements or support strategies can be condensed into the following *abstract goals for operator support*: technologies should facilitate the understanding of system relations, help debias the use and interpretation of data, support the integration of different data, direct attention to non-considered or unavailable information, and foster evaluation and critical thinking. Based on these general support strategies and abstract goals, we specified a number of *functional requirements for technologies*. This was done inductively by clustering our 124 support strategies into requirement categories, while a support strategy could be assigned to one or more categories. This procedure led to the following set of functional requirements:

1. Provide models of the system and connect them with data:
    - Make structural relations and rules explicit;
    - Highlight constraints of the situation and equipment;
    - Enable semantic zooming and switches between levels of abstraction;
    - Process data in a context-dependent manner.

2. Provide and integrate data from different sources:
    - Make data from different sources available;
    - Pre-process and debias data to obtain a valid picture;
    - Make procedures of sampling and integration transparent.

3. Process and integrate data across time:
    - Provide timely information;
    - Enable tracking of changes (past, current, and future data);
    - Link current situation with historical data.

These functional requirements imply that technologies must be able to integrate data from different phases of the plant lifecycle, different levels of the automation pyramid, different parts of the plant, and different points in time. Data integration should be possible even for brownfield plants, because many plants are quite old. The next section provides

an overview of technological concepts and specific technologies that are able to address these requirements.

## 4. Technologies to Support Contextualization

To support contextualization as summarized in the requirements above, information modelling and formal semantics play a central role. In the following sections, we argue that technologies are needed to build and interconnect formal models. These models form the basis for sampling and integrating process data, as well as for connecting the current situation with historical information. Figure 1 provides an overview of the technical concepts and technologies, their links to the functional requirements, and their interconnections.

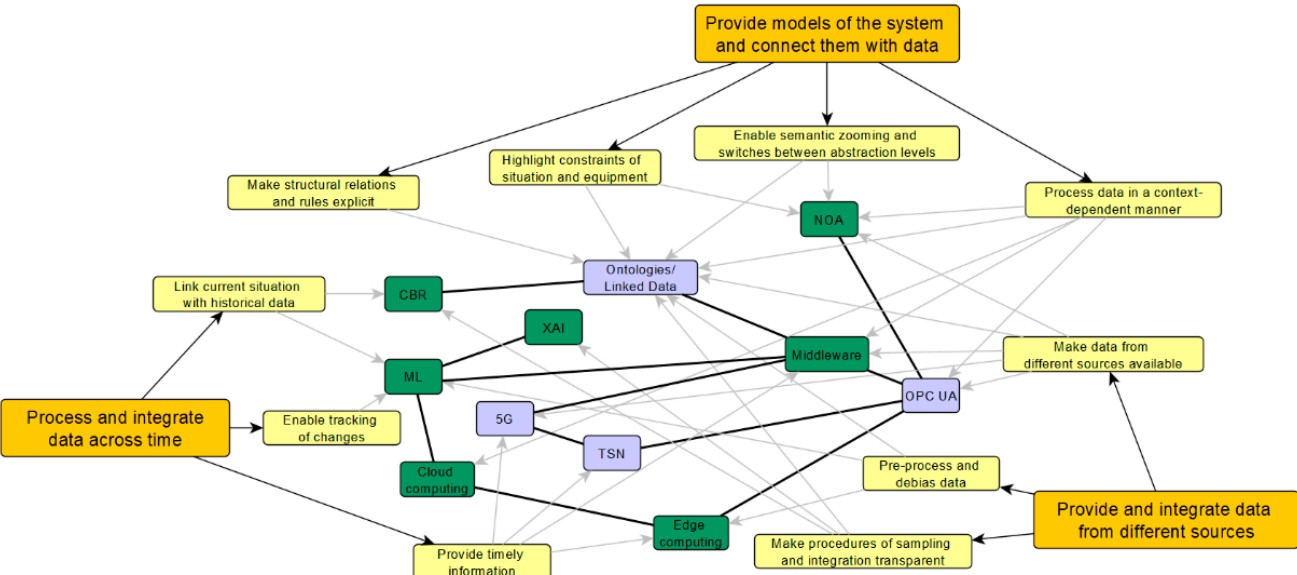

**Figure 1.** Overview of technologies. Yellow boxes represent functional requirements, green boxes represent technical concepts, and purple boxes represent specific technologies. Concepts and technologies are linked to both the functional requirements (grey arrows) and to each other (black bold lines).

Before we delve into the technologies, two clarifications are due. First, the present article does not focus on the contents of formal models. If technologies are to enable an automated use of models, a prerequisite is that these models exist and their contents allow operators to answer the right questions. Therefore, a first step is to understand the details of the production process. This knowledge is often insufficient, either because it is unavailable in principle or because its elicitation from domain experts is a major effort. However, instead of discussing the contents or elicitation methods of formal models, we ask how they can be worked with once they are available. Second, it is important to clarify that most technologies presented in this article are not handled by operators. Instead, they support programmers by enabling an automated use of plant and process knowledge that needed to be looked up and connected manually in the past. Due to the effort associated with that, many promising forms of operator support have not been implemented. Thus, the technologies introduced here do affect operators, but indirectly, by ensuring that the information they need can be provided in a cost-efficient and flexible way.

### 4.1. Building and Interconnecting Formal Models

If technologies are to support operators' understanding of contextual constraints based on the relations in a system, these relations need to be modelled. To this end, an automated integration and use of lifecycle data is needed. This already starts during the *engineering stage* of a plant. First, individual plant equipment must be described. What technical specifications does it have? What can it do? How can it be connected to other

equipment? In the process industries, various exchange formats for this purpose (e.g., DEXPI, NE150, NE159, and NE100) enable different views on a plant for the respective engineering phase [118]. Nevertheless, some important aspects are not formally modelled in contemporary plants (e.g., issues concerning functional requirements). Furthermore, the concept of module type packages (MTP) was proposed for easy integration into higher-level systems to standardize access to process variables and services of autonomous process units [119]. For instance, a module that is capable of tempering and mixing must know about the current state of these functions, and it must communicate this information in a certain manner. Standardized descriptions such as MTP make it easy to integrate the process data of modules from different vendors and provide operators with integrated and consistent information about the module assembly. However, as the interface description does not follow a semantic information model, capability descriptions are not part of the MTP. This requires additional non-standardized documents in the plant design phase to describe services and process variables.

Planning data that are based on formal descriptions such as DEXPI (see above) cannot only describe individual components of a plant but also specify how equipment or functions are *connected*. This requires descriptions on different levels of abstraction such as specific physical connections (e.g., pipe and instrumentation diagrams) or abstract sequences of unit operations (e.g., process flow diagrams). For the definition of single assets (e.g., pump or motor), the concept of an asset administration shell (AAS) was introduced by the Industry 4.0 initiative. This AAS accompanies an asset in all phases of its life, from planning to decommissioning, and keeps all digital information available. Standardized descriptions exist, especially in the form of OPC UA Companion standards (see below) [120].

Such descriptions are not limited to individual machines but should include different production steps within a plant. This knowledge can be used to draw conclusions. For instance, if a buffer is located in front of the fourth of four parallel packaging machines, it follows that problems resulting from the buffer (e.g., warm chocolate due to long storage duration) can only affect the fourth machine but not the three previous ones. As easy as this conclusion is for humans, it must be formalized to make it available for use in operator support systems. This formal modelling can be realized with the help of *ontologies* [121].

However, modelling the physical setup only is a first step, and additional models are needed. In the process industries, the required models are summarized in the *products–processes–resources* (PPR) framework. Process models contain information about the current and future state and behavior of the chemical process, process relations (e.g., thermodynamic, chemical), suitable operation ranges, or safety-related issues. Product models refer to raw materials, intermediates, waste, and end products. They encompass factors such as the physical properties of pure substances, nonlinear interactions in mixtures, product specifications, and risks for human health or the environment. Finally, resource models describe the physical and functional properties of equipment and instrumentation, its behavior, interdependencies between local control units or modules, and requirements for performance and maintenance. The *integration of these partial models* is crucial to formally describe the transformation of substances and the abilities of a plant or production line, which is a prerequisite for operation models that are required to derive optimal process parameters and interventions [122,123]. However, this integration is challenging as the partial models stem from different trades with different perspectives (e.g., machine builders vs. process engineers). Therefore, ontology-based collaboration environments for concurrent process engineering have been developed to facilitate model integration [124], and different ontology-based integration strategies have been compared [121].

How to build the models described above? A promising approach is the use of *Semantic Web Technologies* [125,126]. These concepts are based on graphs (e.g., ontologies) that describe how different elements are connected (for an example, see Figure 2). Graph-based models can be implemented in specific technologies such as Linked Data [127]. To address industrial requirements, the concept of Linked Data has been extended to Linked Enterprise Data [128]. The power of graph structures such as ontologies lies in the fact

that knowledge is interconnected and it is possible to make queries about relations (e.g., What features affect the transport of chocolate bars? What process steps are affected by temperature?). Thus, operators can ask the system to return all elements that have a particular relationship with a concept of interest. To this end, different query languages such as SPARQL [129] are available.

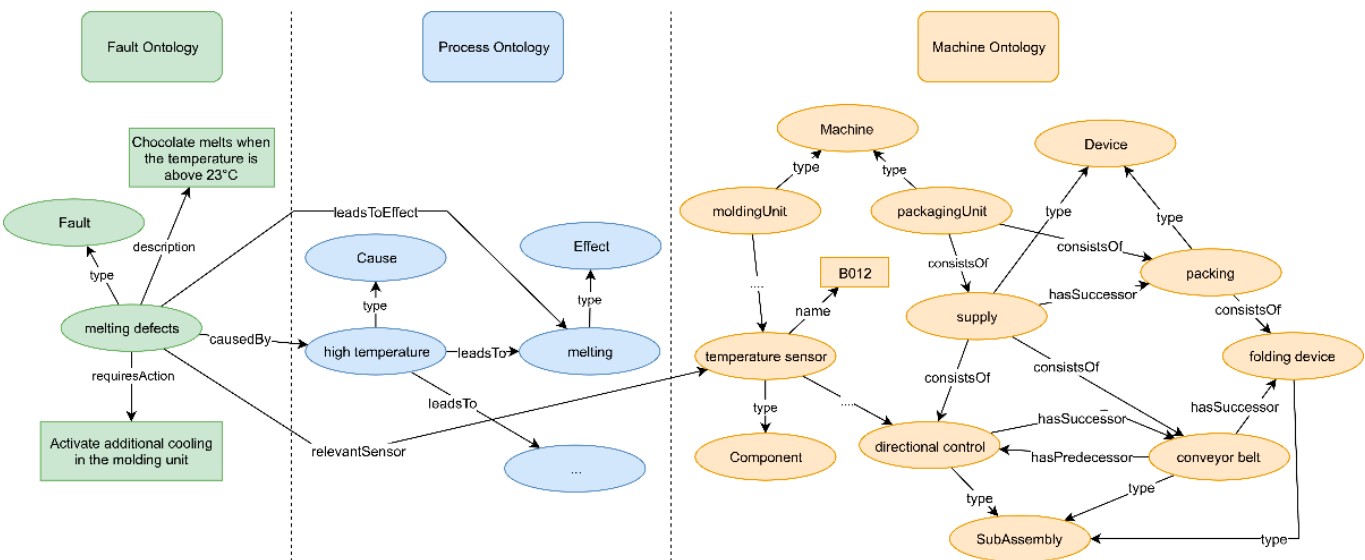

**Figure 2.** Schematic representation of connected ontologies for faults, process, and machines. Oval shapes represent resources (i.e., concepts), rectangles represent literals (i.e., text strings).

Based on an ontological modelling of relationships, knowledge can be generated by drawing inferences from the model (e.g., if chocolate handling operations are affected by bar geometry and the injection into the packing head is a handling operation, the faults of this operation may result from geometrical distortions). Ontologies enable system descriptions on different levels of abstraction (e.g., physics vs. function), can make connections between these levels explicit, and allow operators to switch between them. In this way, ontologies make it possible to inspect the system from different perspectives. Depending on the technology used, they can also make it explicit when information is unavailable or explain why an inference cannot be made. Moreover, different ontologies (e.g., machine and fault ontologies) can be connected to draw conclusions that consider aspects from different knowledge domains. They can also be used to contextualize process data and make them interpretable, which will be explained in the following section.

*4.2. Sampling and Integrating Process Data*

When discussing data sampling and integration, two types of data have to be considered: static and dynamic data. Static data do not change during the production process (e.g., type of machine), whereas dynamic data change more or less rapidly (e.g., temperatures, machine speed, presence of chocolate bars on a conveyor belt). To connect static and dynamic data, semantic descriptions are essential. It should be noted that these descriptions do not suddenly enable an integration of data that has been impossible to date. In fact, data have been integrated with older technologies for decades. However, this had to be done manually for each machine or plant, and thus required immense programming effort. Modern technologies can automate this process, making it feasible to flexibly provide the information needed in a given context. Thus, it becomes realistic to use the potentials of data integration for operator support.

4.2.1. Sampling Dynamic Data

Dynamic data are sampled via different kinds of sensors. Often, not all the relevant data are available, as older plants and machines lack modern sensor technology. In this case, *retrofitting* is possible, e.g., [130] sensors are added that do not intervene in the process but passively collect data and make it available. For instance, a sensor that measures rotation angle can be attached to a chocolate packaging machine and, when combining its data with machine models, it is possible to conclude where each machine part is located at any given time. In this way, potential problems can be eliminated during fault diagnosis (e.g., if a fault occurred while the injector made no contact with the chocolate bar, the injector cannot have caused the broken chocolate). In the process industries, Namur Open Architecture (NOA) provides access to existing field devices and makes it possible to use their information for further applications without interfering with their primary function [131].

However, the present article is less concerned with the question of which data are measured but focuses on the way these data are handled and made available to provide context. This is achieved by *middleware technologies* [132,133]: software that serves as an intermediary between devices and applications. A key technology for handling dynamic data is *OPC UA* [134], a standard that defines how data can be described and exchanged. In addition, the supplementary Companion Specifications create uniform information models that can be used by different manufacturers. In this way, a common understanding of data is guaranteed. The main advantage of OPC UA is that it relies on a semantic information model: devices not only transmit their values but provide information about what they are and how they work (e.g., type of sensor, range of possible values). For instance, this makes it possible to connect a measurement of the current machine temperature with knowledge about where the temperature sensor is located, how it functions, and how temperature affects the process. In this way, knowledge about the plant can be used to interpret the origin and consequences of process data. A disadvantage of OPC UA is its lack of real-time ability. In consequence, it cannot easily handle the fast processes that are common in the discrete processing industry (e.g., packaging 2300 pieces of candy per minute). However, this problem can be solved by combining an OPC UA hardware implementation [135] with time-sensitive networking (TSN) [136]. TSN is a standardized mechanism that is implemented in devices and helps guarantee determinism in network timing, which allows for real-time communication. When combining TSN with OPC UA [137], the advantage of semantics is retained. Although operators usually do not need to see data at such high rates, real-time communication can be a prerequisite for other technologies for operator support (e.g., machine learning that requires data about each chocolate bar).

Computations based on sensor data can be performed at different locations. The decision of where to perform computations depends on where the data are needed and how time-critical they are. On the one extreme, data are processed on servers which are far removed from the process. Such *cloud computing* can be used to integrate data of different sub-plants, enabling conclusions about the state and performance of the entire plant network [138,139]. Moreover, cloud-based services can perform advanced computations on sensor data [140]. In this way, data can be stored and evaluated at the company level across locations. These data can be pure sensor signals or data that were already pre-processed and interpreted. Cloud-based approaches can also be used to distribute knowledge of plants and fault diagnosis to make it accessible to specific groups. The other extreme is decentralized architectures or *edge computing* [141]. That is, sensors and actuators are equipped with local computing power and can perform simple computations themselves. Currently, an increasing amount of smart equipment is being developed [122,142]. Such equipment can also provide information about its own state. For instance, a valve cannot only say whether it is open, but also indicate its degree of fouling. Edge computing can be combined with a hardware implementation of OPC UA to retain the benefits of semantics while using TSN to guarantee high-speed data-processing [135]. Regardless of where the computations are performed, they can provide operators with pre-processed data that are easier to interpret.

### 4.2.2. Integrating Data from Different Sources

To support contextualization, three aspects of integrating data from different sources should be considered: horizontal integration, vertical integration, and the integration of static and dynamic data. *Horizontal integration* means that data from different devices in different parts of the plant are brought together. For instance, the speed of a chocolate packaging machine can be placed in relation to the temperature in a cooler of the previous molding unit. This is a technical challenge, because machines from different vendors cannot easily communicate with each other. Traditionally, this communication has been realized via fieldbus technologies [143]. These standards describe how data are to be handled. An advantage of many fieldbusses is that they can process data in real time. However, a problem is that they have no standardized semantic capabilities. This issue can be tackled with OPC UA, which not only provides an information model (see above) but also a communication standard defining how data are exchanged between different machines. In addition to the cable-based standards, there are first approaches to using the 5th generation mobile radio standard (5G) for industrial applications [144]. This wireless technology makes it possible to distribute data throughout the plant without any need for physical connections between different sensors and actuators. In line with retrofitting and NOA, this enables a use of sensor data beyond what was planned when originally designing the plant. Moreover, 5G enables real-time communication with low latencies and high throughput.

The second aspect of integrating data is *vertical integration*: connecting different levels in the automation pyramid with its field, control, supervisory, planning, and management levels. That is, when interpreting low-level data from a specific sensor (e.g., temperature), it might be necessary to contextualize them with high-level information about production planning (e.g., type of chocolate, current recipe). Vertical integration is supported by the NAMUR Open Architecture (NOA) [145], which describes how data from each level of the automation pyramid can be made available to other levels via open interfaces such as OPC UA. Moreover, integration with data from outside the plant (e.g., forecasts of energy prices) and integration across different agents (e.g., co-workers, customers) might be needed.

A *combination of static and dynamic data* is beneficial for horizontal integration and is the very core of vertical integration. It implies that completely different technologies have to interact with each other. For instance, combining OPC UA with Linked Enterprise Data [146] makes it possible to use the models or ontologies described above to aid in the selection and interpretation of dynamic sensor data. Consider the following fault scenario at a chocolate packaging machine. The fault is known to be connected to melting chocolate (via fault ontologies); it is known that temperatures from the molding unit affect the packaging process (via ontologies of relationships between production steps), and it is known that there is a temperature sensor in the cooler of the molding unit (via plant ontologies). Given this knowledge, it is possible to receive the current temperature from the relevant sensor (via OPC UA or 5G) and relate it to the speed of the packaging machine. Thus, current process data can be selected and interpreted in the light of its relations to data from other production steps. In this way, context can be provided across technology boundaries. Moreover, operators can be informed when data that are important according to the ontologies (e.g., state of packaging material) are unavailable.

### 4.3. Comparing the Present Situation with Historical Data

We have discussed how data can be interpreted based on formal models of the system. However, especially when systems are too complex to model sufficiently and datasets are too large and noisy to be interpretable for humans, an alternative to such model-based integration is to rely on comparisons with the past. Historical data can be used to find patterns and similarities. Such similarities can either be determined based on large sets of process data or based on situation descriptions generated by humans.

### 4.3.1. Finding Patterns in Dynamic Data

A technology that can be used to find patterns and relations in large industrial datasets is *machine learning* (ML) [147]. Based on statistical associations between data, algorithms can make decisions or predictions without being explicitly taught how to do so. In industrial settings, ML has been used for anomaly detection, e.g., [148,149] and fault diagnosis, e.g., [150,151]. Anomaly detection addresses the need to support operators in noticing changes. For instance, based on the sound emitted by ball-bearings, algorithms can determine the degree of wear or damage. This is commonly used for predictive maintenance, e.g., [152], but it can also be helpful during operation. For instance, algorithms could detect when the quality of chocolate bars that exit the molding unit starts to deteriorate. Conversely, classification algorithms for fault diagnosis can inform operators when a current situation is similar to a previously encountered fault. For instance, motor current timelines might reveal why a tray packer that puts chocolate packages into boxes produces a fault, because different fault types lead to characteristic changes in these curves. Such changes might be barely detectable for humans when visually inspecting the curves, but machine learning is able to find small effects in noisy datasets.

A major problem of machine learning is that it is hard to understand what the algorithms are doing. To address this lack of transparency, the concept of *explainable artificial intelligence* (XAI) has been put forward [153,154]. For instance, attention mechanisms can indicate what information algorithms have used to generate a solution [155]. Thus, when analyzing images of chocolate exiting the molding unit, the algorithm could indicate what areas of the chocolate it has looked at to determine that its quality is insufficient. Four types of XAI technique can be distinguished [153]. Visualization techniques show model representations to reveal the patterns inside neural units (e.g., tree view visualizations). Knowledge extraction techniques try to extract the knowledge a model has acquired during training. Influence methods estimate how important a feature is for the model by changing this feature and observing its effects on performance. Finally, example-based explanations reveal how a model performs with particular instances of a dataset. Research on the effects of XAI has mainly focused on its potential to enhance trust and compliance with the recommendations provided by algorithms [156,157]. However, explanations can also foster the generation of mental models about the algorithms [158] and support users in identifying false solutions, leading them to cross-check the ML results more carefully [159].

### 4.3.2. Storing and Retrieving Human Experience

Another valuable source of historical information operator experience, its storage and retrieval can be supported by *case-based reasoning* (CBR) [160]. Based on a situation description by the operator, the system searches for similar cases in a database. An advantage of CBR is that it is explainable [161]. Thus, the system can make its selection of cases transparent, justify its sampling strategies, explain the relevance of questions, help users understand the meaning of concepts, and support learning [162].

Traditional CBR systems rely on a match between specific features. For instance, if operators describe the current situation by stating that a high temperature in the production hall and broken chocolate at the carrier belt are present, the system retrieves cases that share the exact same features. To benefit from knowledge of system relations, CBR can be combined with ontologies [163,164]. This makes it possible to go beyond specific features and retrieve cases that share an underlying causal principle (i.e., high temperature causes conveyor soiling, conveyor soiling causes displacements of chocolate bars, and displacements cause breaking under high mechanical impact). In this way, cases can be retrieved that differ in terms of their specific features but might still afford the same solution (e.g., increase machine cooling or reduce machine speed). Moreover, ontological modelling enables systems to make such matches between structural relations explicit. In this way, operators can be supported in transferring knowledge to situations that differ in terms of their surface features. However, a major technical challenge is that this requires a

system that can interpret ontologies and make appropriate queries to extract the relevant information.

CBR can be combined with dynamic process data, and some CBR systems are capable of autonomous information gathering [165,166]. They retrieve information from databases so that users do not have to describe all features of the situation by themselves. However, enabling systems to retrieve relevant process data requires a combination of CBR with the technologies described above. Based on formal models of the technical system, a CBR system would know where to look for relevant information and, based on OPC UA, it could extract this information from the respective machines. Such automatic extraction of relevant process data would allow for much richer case descriptions, enabling future users to assess whether the context of a previous situation matches the current one.

## 5. Discussion

Contextualizing data and observations is a major challenge for operators of industrial plants. The present article addressed this challenge by presenting examples of the psychological literature that deals with the questions how people sample information, integrate different information elements, categorize objects and events, and reason about causes. In each area, characteristic limitations and biases were reported, which should form the basis for the conceptualization of support strategies. Based on the literature section, we extracted three groups of functional requirements for digital transformation technologies: they should connect data to models of the system, provide and integrate data from different sources, and process and integrate data over time.

A variety of technologies can address these requirements and support contextualization. At the core of this endeavor is the generation and combination of formal models of physical and functional system relations. This can be achieved by Semantic Web technologies such as ontologies and Linked Data. These models can provide the basis for information sampling and integration. A model-based integration of data from different sources calls for standards such as OPC UA that provide semantic modelling capabilities and can be combined with other technologies, for instance to enable real-time communication. Machine learning can find patterns in complex datasets by matching dynamic and historical data, but an important requirement is that it should be explainable. Moreover, operator experience can be made available via case-based reasoning. Again, such approaches would greatly benefit from a combination with formal models of system relations. Thus, the integration of different technologies provides immense potential to support contextualization.

### 5.1. What Stands in the Way of Application?

Most technologies presented in this article are not completely new. This raises the question why contextualization still is a challenge. Why are the available technologies not used as much as they could be? Several reasons might account for this. First, using the technologies can be *difficult and effortful*. For instance, eliciting high-quality models of system relations from domain experts is a major effort [167]. In complex industrial systems, models are incomplete and not even engineers can fully specify them [168,169]. For instance, it is impossible to exhaustively describe the behavior of chocolate under all conditions that might occur in a plant. The modelling of system relations gets even more difficult when models have to bridge domain boundaries (e.g., interactions of chocolate characteristics and machines). Some people hope that purely data-driven approaches such as machine learning can compensate for a limited understanding of the underlying processes. However, statistical associations are not always informative. A promising alternative is to use causal models that combine data based on relatively simple causal diagrams by performing particular computations [98]. Initial evidence suggests that such models can produce better predictions than searching for statistical associations and controlling for known covariates [170]. Concerns about feasibility also apply to purely data-driven approaches. If data are to be integrated, they must be available in the first place.

However, lots of important data (e.g., state of packaging material) are not measurable in principle, in-line, or at a reasonable price [171]. Additionally, simply acquiring more data is insufficient if their quality is not guaranteed, as data can have biases, ambiguities, and inaccuracies [172].

A second reason that people do not use the full potentials of digital transformation technologies is that they are *not used to it*. For instance, despite the invaluable power of the semantics offered by OPC UA, current controllers do not provide sematic information models [173]. They do use OPC UA, but in the same way as they have previously used list-based fieldbus standards. Concepts and technologies that are established in other domains (e.g., ontologies from the area of knowledge engineering) are only gradually being adopted, and it takes time for industrial practitioners to recognize their potential.

A third, similar reason is that engineers often *do not know how to apply* the technologies in ways that bring out their benefits for human–machine interaction. For instance, it is unclear how to use ontologies in ways that help people think (e.g., encourage them to consider alternative hypotheses). Psychological knowledge is not usually considered in the industrial application of technologies. Moreover, a psychologically valuable application of ontologies calls for immense formalization efforts and sophisticated querying mechanisms. Therefore, applying digital transformation technologies to optimize human–machine interaction will require two things: technological advances to support the people who conceptualize and implement these technologies, and a close cooperation between different disciplines such as computer scientists, engineers, and psychologists.

*5.2. The Question of Function Allocation*

Contextualization is not only beneficial for humans but also for technical systems. For instance, it can be used for automatic failure detection and prevention [15]. Thus, the technologies described in the present article can be used to increase the level of automation, enhance machine-to-machine communication, and minimize the role of operators. In fact, keeping humans out of the loop is what many technological developments strive towards. For instance, machine learning often confronts operators with a decision but does not incorporate them in the evaluation and integration of the data that led to this decision. On the one hand, one might argue that this is a good thing, as algorithmic integration is often superior to human integration [55,56] and the large amount of data in complex industrial systems makes automation indispensable for information processing and integration. On the other hand, suggesting specific decisions can foster an uncritical acceptance of these decisions, without cross-checking them [18,40], and these negative effects increase with the degree of automation [174]. Although the ironies of automation were described decades ago [175], they have not made their way into the design of many real-life applications.

A related question is what information should be available to operators to provide them with the necessary tools and knowledge to intervene if necessary. This question presents another dilemma. On the one hand, human–machine cooperation requires transparency [176–178]. When humans only receive highly constrained information, they cannot form an accurate mental model of the system. From this perspective, it might be necessary to share more information with operators instead of only making it available for internal use by the technologies. On the other hand, information overload can decrease the quality of decisions [46]. Therefore, how to provide information in ways that do not overload operators but foster productive thinking remains an important issue [5].

Such changes, enabled by a context-aware, interconnected production system, are likely to generate benefits not only from the perspective of the production process, but also from the perspective of the human operators themselves. By allowing them to derive a more complete understanding of the technical system and the current situation, contextualization can create more meaningful jobs [179] and lead to operator empowerment [180,181]. The resulting increases in autonomy and flexibility are likely to enhance motivation [182] and foster lifelong learning [22].

The changes in function allocation between humans and machines that typically accompany technological advances have consequences far beyond the expected [183]. Imagine an ideal world, in which technologies are used in the most thoughtful, human-centered way: operators can inspect overviews of system relations, which are complemented with relevant and up-to date context information from different sources. Will this lead to new problems? If the presented system relations appear to be conclusive and complete, this might encourage operators to trust them and fail to notice mistakes. Similarly, explainable artificial intelligence (XAI) can lead people to over-rely on the system and comply with false explanations [156,184,185]. We must consider that each new technology will change the situation in unexpected ways. Therefore, it is crucial to keep people thinking. Addressing even unexpected reasoning biases may require approaches that have not been considered in human–machine interactions to date. For instance, according to the argumentative theory of reasoning [186], a powerful way to eliminate reasoning biases is to put people into an argumentative context in which they have to disconfirm the reasoning of others. Should we make operators argue with the technologies to achieve the best cooperative performance? Obviously, ideas that are based on research on human cognition must be balanced with what is feasible in a production context.

### 5.3. Limitations and Future Work

### 5.3.1. The Psychological Literature Does Not Always Adequately Address Real-World Demands

We mainly derived our contextualization challenges and our requirements for technologies from basic research in cognitive psychology. As the tasks used in these studies are much simpler than the operation of industrial plants, it is unclear to what degree the results are applicable in more complex environments. An alternative approach is to derive requirements from on-site observations in the domain. Although this would certainly increase ecological validity, it also has pitfalls. It would only be possible to assess the status quo (i.e., how operators work today) and it is hard to draw conclusions about counterfactuals (i.e., what would happen if different technologies were available). People cannot provide valid reports of their cognitive processes and limitations. Moreover, they often cannot imagine what would help them when it is not yet available. Therefore, a mere focus on domain observations is insufficient, and a multi-method approach is required that also takes psychological knowledge about cognitive processes and biases into account.

It also must be considered that the introduction of cyber-physical production systems (CPPS) will pose its own challenges for operator work [22,23,179]. One exemplary challenge is that the requirement of being 'in control' becomes more abstract [20]. For instance, consider anticipation. In traditional supervisory control, operators had to anticipate trends of process parameters, while in CPPS they also have to anticipate when controllers will become unable to self-regulate or the system will exhaust its adaptive capacities. Even in today's systems, people do not sufficiently understand the function of automatic controllers or the conditions that make control non-effective [187]. With more system autonomy, this challenge will be exacerbated and it will be necessary to investigate the cognitive demands specific to this new workplace.

### 5.3.2. Only a Fraction of the Relevant Cognitive Challenges Was Addressed

In the present article, we considered four psychological areas, but it is impossible to provide a comprehensive review of these areas in just one journal article. Moreover, the selection of areas is limited as well, and we did not address all cognitive challenges of contextualizing data in complex systems. Dörner (as cited in [188]) postulates four process components of complex problem solving: (1) gathering information about the system and integrating it into one's model of the system, (2) goal elaboration and goal balancing, (3) planning measures and making decisions, and (4) self-management. We addressed the first component, asking how people sample and process information, but we did not consider how people subsequently use this information. Digital transformation

technologies certainly have the potential to also support the other three components of problem solving, although different technological support strategies may be helpful (e.g., simulations). Future work should specify the cognitive challenges and technological potentials of contextualization from a more action-oriented perspective, and draw on the rich knowledge base of complex problem solving research (for an overview, see [189]).

Several other relevant psychological issues remained unaddressed. For instance, social factors can affect contextualization, and research into issues such as social influence [190], team coordination and reasoning [191], or team situation awareness [192] provides valuable insights that should be considered when designing digital transformation technologies. Additionally, information processing is subject to interindividual differences. One such difference concerns people's knowledge: the information search depends on prior knowledge [193], and while experts do not always use more cues than novices, they use more relevant ones [194]. Moreover, behavioral habits influence the search for and utilization of information. People with strong habits often refrain from acquiring information but simply select the option they usually select, irrespective of contextual constraints [195]. Thus, there are many more issues worth considering.

### 5.3.3. Support Strategies from the Psychological Literature Were Not Considered

The present article focused on information modelling technologies and largely ignored the question of how information should be presented in the human–machine interface. This focus was chosen because much of the Human Factors' literature is concerned with interface design, while information modelling has not received much attention to date. Concerning the question of how to design interfaces that support contextualization, promising insights can be drawn from the work on ecological interface design [6,8,196], visualizations of causal relations [197,198], and visualizations that provide continuity and context across interfaces with different levels of abstraction [199]. Besides interface design, the psychological literature provides an arsenal of methods to support contextual and causal reasoning via instruction and training. For instance, overconfidence can be reduced by asking people to consider the unknowns [43], the understanding of causal relations can be enhanced by explication and structural alignment [200], and the learning and transfer of complex system principles can be supported by simulations [201]. Future work should extract support strategies from different fields of psychology and integrate them with the technological possibilities of digital transformation.

### 5.3.4. No Specific Implementations of Technologies Were Suggested

The article aimed to serve as a starting point for an interdisciplinary cooperation between engineers and psychologists on the design of CPPS. Accordingly, we provided a broad overview of different psychological issues and technologies, instead of going into depth for any of them. A useful next step would be to translate scientific knowledge about one particular psychological issue into detailed requirements for one specific technology, and vice versa. For instance, if you want to foster the consideration and evaluation of alternative explanations in causal reasoning, what does this mean for the design of ontologies and the application of query languages? If you are developing a CBR system, what would this system need to do to avoid sampling biases? We hope that the present article will help to generate many such questions, which can then be pursued in subsequent interdisciplinary work.

### 5.4. Conclusions

Contextualization is a major challenge in the process industries and discrete processing industries. It requires operators to sample the correct information, integrate it in suitable ways, appropriately categorize situations, and draw valid inferences about causal relations. These cognitive activities can be supported by digital transformation technologies that provide formal models of the technical system, rely on these models to access and integrate data from different sources, and use historical data to interpret current situations. Based on

these technologies, relevant context information can be made accessible to human operators. For instance, the functional and causal relations within the system can be made explicit, the interactions between data from different sources can be visualized, or the results of machine learning algorithms can be explained. To leverage the full potential of these technologies, their integration needs to span technological and disciplinary boundaries. For instance, operator experience with regard to specific fault situations can be mapped onto formal models of system relations.

In principle, the digital transformation technologies that support contextualization could be used in two ways: either to further exclude humans from production processes, or to finally establish a genuine human–machine cooperation that results in the optimization of the entire system. The human-centered application of these technologies faces a number of challenges: Their application can be difficult and effortful, engineers are not yet used to the new potentials, and they often lack the psychological knowledge necessary to apply the technologies in the way that is most conducive to human reasoning. Thus, a close interdisciplinary cooperation will provide fruitful approaches to answering the question of how digital transformation can support human reasoning in CPPS.

**Author Contributions:** Conceptualization, R.M.; methodology, R.M., F.K., J.R., D.W.H.; formal analysis, R.M., F.K.; investigation, R.M., F.K., J.R.; writing—original draft preparation, R.M., F.K., J.R.; writing—review and editing, F.K., J.R., D.W.H.; visualization, R.M., J.R.; supervision, R.M.; project administration, R.M.; funding acquisition, R.M. All authors have read and agreed to the published version of the manuscript.

**Funding:** This research was funded by the German Federal Ministry of Education and Research (BMBF), grant number 02K16C070, and the German Research Foundation (DFG), grant number GRK 2323/1 and grant number PA 1232/12-3.

**Data Availability Statement:** Not Applicable, the study does not report any data.

**Acknowledgments:** We want to thank Annerose Braune for helpful comments on an earlier version of the manuscript.

**Conflicts of Interest:** The authors declare no conflict of interest. The funders had no role in the design of the study; in the collection, analyses, or interpretation of data; in the writing of the manuscript, or in the decision to publish the results.

## Appendix A

The following tables translate the cognitive factors identified in Section 2 into general requirements for operator support strategies, and present examples from a chocolate packaging scenario to illustrate these requirements. They are organized according to the four areas reviewed in Section 2: sampling the available information (Table A1), integrating different information elements (Table A2), categorizing objects and events (Table A3), and reasoning about causes (Table A4). When providing examples in the third column, plain text indicates general instructions for operator support, whereas text in inverted commas represents the information content that could be communicated to operators, without intending to implicate that the exact formulation should be adopted.

**Table A1.** Sampling the available information.

| Cognitive Factors | General Requirements for Operator Support Strategies | Examples |
|---|---|---|
| | Information samples are biased | |
| Sampling biases | Reduce bias in the automated generation of samples | Do not perform product control at fixed intervals (e.g., every 100th casting mold) as they might coincide with temperature cycles |
| | Make the selection of automatically generated samples transparent | "Only every 100th casting mold is submitted to quality control" |

**Table A1.** *Cont.*

| Cognitive Factors | General Requirements for Operator Support Strategies | Examples |
|---|---|---|
| | Provide information as to whether samples are representative | "The frequency of this fault have only been determined for milk chocolate but not for caramel chocolate" |
| | Provide hints when operators have ignored potentially relevant information | "You have not checked motor current trend charts, yet" |
| | Support the generation of arguments against a given anchor or standard | "Please check whether the temperature value from the selected previous case applies to the current situation" |
| Availability heuristic | Make information available in an unbiased manner | Provide different data sources (e.g., temperature in end cooler, buffer, and packaging machine) and data types (e.g., temperature, soiling, and motor currents) |
| | Provide information about the base rates of process states and events | "Hollow bottoms are present in 20% of the bars overall" |
| | Provide hints that other information sources are available | "The current hypothesis can also be checked by inspecting the motor current trend chart" |
| Conditional sampling | Make it explicit when events may be overrepresented in samples | "Chocolate bars have mostly been checked for hollow bottoms when faults have occurred, which may overestimate the impact of this deviation" |
| | Use presentation formats that aid Bayes reasoning | Frequency grids or frequency trees |
| Repeating choices that initially led to good outcomes | Provide hints about own previous choices and choices of others in similar situations | "In previous instances of this fault, you have only checked for hollow bottoms. Other operators have also checked motor current trend charts and vacuum suction" |
| | Make the relevance of different data sources and observations transparent | "Bar skewness after the stopper is predictive of this fault, while bar skewness before the stopper is not" |
| | Make it explicit when the contribution of data sources is context-dependent | "Room temperature is predictive of this fault for milk chocolate but not dark chocolate" |
| | Selecting and ignoring particular types of information | |
| Salient cues | Make cue validities explicit | "Hollow bottoms have low predictive value for this fault" |
| | Support evaluations of whether extreme values are relevant | "High room temperature is irrelevant for bad packaging quality" |
| | Make non-salient but relevant cues accessible | Provide height measurement of chocolate bars, which can cause problems but is not visually perceivable for operators |
| Confirmation bias | Provide evidence/data in favor of opposing hypotheses | "You assume the cause for skewed bars to be insufficient ground contact due to hollow bottoms, but the motor current trend chart indicates too much grip due to a smeared conveyor belt" |
| | Make it transparent which hypotheses are supported by what information and which have not been tested sufficiently | "The hypothesis of insufficient ground contact as a cause for skewed bars is supported by hollow bottoms, but the motor current trend chart indicates a soiled conveyor belt" |
| Positive test strategy | Present data for situations in which the property of interest differs, thus mitigating the effect of illusory correlations | "Bars had hollow bottoms in 25% of the situations in which this fault occurred, but also in 18% of situations without a fault" |
| | Make unequal sample sizes transparent | "There are 311 measurements for milk chocolate but only 17 for caramel chocolate" |
| | Weight evidence by sample size | "The 17 measurements for caramel chocolate may not be representative" |

**Table A1.** *Cont.*

| Cognitive Factors | General Requirements for Operator Support Strategies | Examples |
|---|---|---|
| Re-interpreting information to fit hypotheses | Ask operators to make interpretations explicit, provide feedback about these interpretations | "What do you conclude from the observation that bars were skewed?" [Operator: "Insufficient ground contact due to hollow bottoms"] "This conclusion is problematic, because bar skewness can also be caused by too much grip on a smeared conveyor belt" |
| Inattentional blindness | Direct operators' attention to information they have not yet considered | "This problem can also be checked by inspecting the motor current trend chart" |
| Ignoring contextual constraints | Make constraints and side-effects transparent | "Reducing machine speed below 500 will cause overflow in the buffer, which may ultimately force the molding unit to stop" |
| Neglecting unknown or missing information | Make it explicit that data for some relevant aspects are not available | "Temperature of chocolate filling affects bar stability but cannot be measured" |
| | Highlight the consequences of missing information (what-if) | "If the (unavailable) temperature of chocolate filling strongly differs from temperature of the coating, this can lead to tensions and cause breakage" |
| Tasks and information sources affect information search | | |
| Task complexity | Provide information in a task-/state-/context-dependent manner | "To determine whether the problem may have been caused by bar temperature, you also need to consider the temperature in the molding unit and whether this chocolate type is susceptible to temperature variations" (and provide specific values) |
| | Make additional information available on demand | "Click here to check information about the machine, from the molding unit, and from the production planning system …" |
| | Provide domain information and problem solving information (instead of just basic data) for complex tasks | "Too low temperatures of the cold stamp lead to ice crystals, which cause instable walls" instead of just "The temperature in the end cooler is −10 °C" |
| Information overload | Filter data according to their relevance, use context-dependent filtering | Do not show foil characteristics in case of problems at the feed conveyor |
| | Highlight the relevance of information elements | "Temperature variations are particularly important for this fault" |
| | Make consequences of misuse understandable (e.g., consequences of errors and filtering) | "If you ignore motor current trend charts, you may not find out whether this problem is caused by too much grip on the conveyor belt" |
| | Make information sources explicit | "Too much grip on the conveyor belt can be estimated from motor current trend charts" |
| | Provide information on different levels of abstraction and support switching | "Physical Form: hollow bottoms; Physical Function: insufficient ground contact on conveyor belt; Generalized Function: dysfunctional transport of chocolate bars; Abstract Function: frequent faults at carrier belt; Functional Purpose: low efficiency" |
| Accessibility | Increase accessibility of context information (e.g., right format, right level of detail, saving time, lots of information in one place) | Name and explain current problems in the molding unit (previous production step), instead of individual molding parameter values |
| | Make format and level of detail configurable | Let operators decide whether they want to see overall efficiency during the previous shift or individual faults |
| | Integrate information from different sources | Present information about chocolate characteristics and state of the molding unit together with associated problems of the packaging machine |

**Table A1.** *Cont.*

| Cognitive Factors | General Requirements for Operator Support Strategies | Examples |
|---|---|---|
| Familiarity | Make unfamiliar sources easy to access | Place a link to motor current trend charts on the main fault screen |
| | Highlight consequences of only considering particular (familiar) sources | "If you only look at bar skewness but not motor current trend charts, you may not see whether the problem is caused by too much grip on the conveyor belt" |

**Table A2.** Integrating different information elements.

| Cognitive Factors | General Requirements for Operator Support Strategies | Examples |
|---|---|---|
| | Strategies of information integration | |
| Formal or informal | Integrate cues algorithmically, especially in low and high-validity situations | Automatically calculate whether speed of molding unit matches the speed of packaging machines |
| | Make algorithms transparent (i.e., use of data, algorithmic procedures), allowing operators to evaluate completeness and appropriateness | "The image processing algorithm has attended to the surface of the casting mold when determining that the mold was faulty" |
| | Enable operators to include/exclude cues and change weights, show changes in outcomes | Operators can indicate that temperature is less important in a particular situation, and in consequence it receives less weight in the selection of similar cases |
| | Make it explicit what important factors an algorithm cannot consider | "The algorithm ignores temperature of the filling as it cannot be measured, and air moisture as its specific effects are unknown" |
| Rule- or exemplar-based | Make appropriate rules available, depending on task, goals, and context | "High temperature may predict soiling of the conveyor belt, but more so for milk chocolate than dark chocolate" |
| | Support cue abstraction (i.e., make predictive power of cues explicit) | "Hollow bottoms are the strongest predictor of skewed chocolate bars" |
| | Present each cue in terms of its presence/absence or value | "Hollow bottoms are present, temperature is 21 °C" |
| | Support selection of suitable exemplars (cases) | "Case 12 matches the current situation in terms of temperature, motor currents, and chocolate type, Case 17 differs in chocolate type" |
| | Highlight correspondence between cases and rules (i.e., case abstraction) | "Case 14 has a lower molding temperature but still is comparable as the bars remained in the buffer for longer, which also leads to warming" |
| Relying on heuristics | Support operators in assessing the context-dependent suitability of heuristics | "To determine whether there was a problem with a packing claw, it is sufficient to check whether every eighth bar was affected" |
| | Point out heuristics that lead to problematic outcomes | "Evaluating the height of the downholder in isolation is problematic, because a low downholder can be suitable if bars are smaller" |
| | Reduce the need to rely on heuristics by providing algorithmic integration | Automatically calculate whether speed of molding unit matches the speed of packaging machines |
| | Task and information characteristics affect integration | |

**Table A2.** *Cont.*

| Cognitive Factors | General Requirements for Operator Support Strategies | Examples |
|---|---|---|
| Task complexity | Support operators in partitioning the task | "(1) Check where the problem starts occurring, (2) check whether all bars or only some of them are affected, (3) check bar characteristics like geometry and temperature, (4) check mechanical machine settings" |
| | Support data integration and provide overviews in complex tasks | Present information about chocolate characteristics and state of the molding unit together with associated problems of the packaging machine |
| Coherence | Highlight whether available information is coherent (i.e., points in the same direction) and point out mismatches | "Hollow bottoms are consistent with insufficient ground contact, but the motor current trend chart indicates a soiled conveyor belt" |
| | Organize information to provide overview and facilitate assessment of coherence | Present current values of all variables that affect chocolate smearing in one place (even when they stem from different production steps) |
| Validity of easily accessible information | Make valid/important information easy to access | Present the five parameters that best predict the current fault type on the main fault screen |
| | Make validity of information transparent | "Hollow bottoms have low predictive value for this fault" |
| Presentation format | Reduce search demands by integrating information from different sources | Present information about chocolate characteristics and state of the molding unit together with associated problems of the packaging machine |
| Time pressure | Provide higher degree of automated integration in situations with high time pressure | Automatically integrate high temperature in molding unit, long time in buffer, high motor currents, and milk chocolate into "the conveyor belt may need to be cleaned" |

**Table A3.** Categorizing objects and events.

| Cognitive Factors | General Requirements for Operator Support Strategies | Examples |
|---|---|---|
| Differentiation: context-dependent, flexible categorization | | |
| Flexible category representations | Make interconnections of information transparent | "Both long times in the buffer and high temperatures may cause melting, but what is an appropriate temperature depends on chocolate type"; "bar weight often provides information about bar height" |
| | Allow operators to access information from different perspectives (e.g., levels of abstraction, task-dependent views) | "Physical Form: hollow bottoms; Physical Function: insufficient ground contact on conveyor belt; Generalized Function: dysfunctional transport of chocolate bars; Abstract Function: frequent faults at carrier belt; Functional Purpose: low efficiency" |
| | Provide different instead of just similar cases | "Case 26 has a similar symptom (soiled conveyor belt) but different parameter values, and should be cross-checked as a differential diagnosis" |
| | Facilitate information decomposition | "Conveyor belt soiling should be checked for all four conveyor belts, because if it only occurs at the vacuum belt, this may indicate wear of the vacuum hole edges" |
| | Suggest different interpretations or prompt operators to generate them | "A soiled conveyor belt can indicate a temperature problem, but it can also indicate friction due to differential speed of adjacent conveyors" |

**Table A3.** *Cont.*

| Cognitive Factors | General Requirements for Operator Support Strategies | Examples |
|---|---|---|
| Environmental contingencies | Provide information about relations and parameter interactions | "Temperature in the molding unit and time in the buffer both lead to warm chocolate, which can cause soiling. However, the two parameters can compensate each other" |
| | Highlight changes in regularities of events | "High temperature is less problematic today, because dark chocolate is being produced" |
| | Highlight changes in relations between interacting parameters | "High temperature is less problematic now, because you have reduced machine speed" |
| Conceptual change | Provide factual information to help operators detect and correct false beliefs | "The turning wheel is not responsible for squished chocolate bars" |
| | Provide information about system structure and functional relations to support mental model generation and updating | "Hollow bottoms cause problems because they reduce ground contact on the conveyor belt, which may lead to misalignment of chocolate bars" |
| | Provide semantic relations between concepts and events to make ontological structures understandable | "Temperature in the molding unit and time in the buffer both lead to warm chocolate, which can cause soiling. However, the two parameters can compensate each other" |
| Cue relevance | Highlight and explain context-specific changes in the relevance of information | "If machine speed is reduced, hollow bottoms are less problematic, as the bars are not pulled into a skewed position at conveyor boundaries as much" |
| Generalization: the role of similarity | | |
| Similarity depends on context | Highlight the features most important to determine similarity in the current context | "With milk chocolate, temperature the most important parameter to determine whether a previous case is similar" |
| Similarity depends on focus of attention | Focus attention on relevant features depending on context | "If you want to find out whether soiling of conveyor belts may have caused the problem, you should focus on parameters that are associated with melting: temperature and time in the buffer" |
| Surface/structural features | Show structural correspondence between situations | "Case 14 has a lower molding temperature but still is comparable as the bars remained in the buffer for longer, which also leads to warming" |
| | Provide information as relational categories rather than just entity categories | "Chocolate types that melt easily" and "chocolate types that break easily", rather than just "milk chocolate" and "marzipan" |
| | Highlight differences despite feature similarity | "In Case 13, machine speed and temperature were as high as in the current situation. However, as dark chocolate was produced, these parameters were suitable, while now they may be problematic" |
| Features beyond similarity | Make it explicit when comparisons should change depending on task goals | "If you need to solve the problem quickly, select cases that offer solutions relying on machine speed or cleaning instead of cases that require changes in the molding unit" |
| | Support comparison between situations according to different criteria, let operators manipulate these criteria | Offer case similarity calculation methods based on feature similarity, mechanism similarity, outcome quality, or required effort |
| | Show information distribution within situation classes (e.g., variability of parameters) | "For this fault, temperature fluctuations are normal: temperature is very high in some cases of this fault class but not in others" |

**Table A4.** Reasoning about causes.

| Cognitive Factors | General Requirements for Operator Support Strategies | Examples |
| --- | --- | --- |
| | Covariation | |
| Magnitude of probabilities | Make base rates of causes and outcomes available | "Hollow bottoms occur in 20% of situations, but the problem of skewed bars after the stopper only occurs in 5%" |
| | Point out illusory correlations | "This fault type occurs almost as often when hollow bottoms are present and when they are absent" |
| | Make absence of causal relations explicit | "Foil color differs between the current situation and the selected case but has no impact on the current problem" |
| Inattentional blindness for negative relations | Make negative causal relations explicit | "Machine cooling reduces the impact of long time in the buffer" |
| Overshadowing and super-learning | Disentangle single causal effects | "Skewed bars can be caused by soiled conveyor belts, hollow bottoms, insufficient vacuum suction, and incorrect positioning of machine parts" |
| | Make interactions between causes explicit (e.g., additive, enhancing, suppressing) | "The effects of low temperatures in the molding unit are cancelled out when chocolate bars remain in the buffer for a long time" |
| | Highlight simple correlations that do not contribute to causal effects | "Low weight of chocolate bars is associated with skewness, but this is not because weight causes skewness but because weight and skewness both are a consequence of hollow bottoms" |
| Type of task | State alternative outcomes in predictive reasoning tasks | "Low temperatures of chocolate filling may not only cause hollow bottoms but also tensions between coating and filling, which can lead to breakage" |
| | Highlight causal strengths in predictive reasoning tasks | "Low machine cooling only has a weak impact on conveyor belt soiling" |
| | Temporal relations | |
| Temporal order | Provide information about temporal order and temporal dependencies of events | "First, high temperatures cause chocolate to melt and lead to soiling of the conveyor belts. This can result in skewed bars, which later may break upon contact with the carrier belt" |
| | Highlight temporal orders when they cannot easily be perceived | "A chocolate bar is first touched by the injector and then by the transfer finger, but it can quickly move back and forth between the two components, which may cause breakage" |
| | Make it explicit when events follow each other but are not causally related | "Bars can break after they have moved into the turning wheel, but it is not the turning wheel that causes breakage" |
| Temporal contiguity | Minimize time delays in information presentation | Present past molding problems together with current packaging problems instead of at the time when they occur (about one hour delay) |
| | Increase time delays when no causal relation exists | Present the processes in the injection unit separately from processes in the turning wheel when explaining how the former cause breakage |
| Temporal variability | Make variable time lags between cause and effect transparent | "Problems in the molding unit can affect the packaging machine with a delay of 20 min to 1.5 h" |
| | Provide information about factors affecting the variability of delays | "The delay of molding problems affecting the packaging machine depends on the time that chocolate bars remain in the buffer" |
| Expecting longer delays | Provide information about delayed effects | Simulate how the current temperatures in the molding unit will affect the packaging process in one hour |
| | Prior knowledge | |

**Table A4.** *Cont.*

| Cognitive Factors | General Requirements for Operator Support Strategies | Examples |
|---|---|---|
| Assumptions about causal roles of events | Make causal relations within the system and the causal roles of each factor transparent | "Temperature in the molding unit and time in the buffer both lead to warm chocolate, which can cause soiling. However, the two parameters can compensate each other" |
| | Point out common misconceptions for a given problem situation | "People often think that the turning wheel squishes the bars, but that is not true. The problem actually originates in the injection unit and only becomes visible at the turning wheel" |
| Selecting appropriate integration rules | Support integration of fragments of causal nets | Show causal diagram of factors from different process steps affecting the breakability of chocolate bars |
| | Make relations between causes explicit (e.g., additive, compensatory) | "Bar height and downholder height can compensate each other" |
| | Show how relations are affected by context, highlight differences between situations | "Whether high temperatures cause skewed chocolate bars depends on machine speed, and the relation is stronger for milk chocolate than dark chocolate" |
| Experience | Specify the exact types of relations between events or elements of the plant instead of just their causal direction | Show mechanisms by which high temperatures cause skewed chocolate bars (soiling of conveyor belts, increasing grip on the belts, increasing the effects of speed differences at conveyor boundaries) |
| | Support novices by highlighting causal phenomena (e.g., negative feedback) | "Reducing machine speed can mitigate problems due to warm chocolate, but it also increases time in buffer, which can cause even more warming" |
| Dealing with complexity | | |
| Linear reasoning | Make non-linear interactions and complex system features transparent (e.g., emergence) | Show how problems result from interactions of temperature, speed, machine and chocolate characteristics, and mechanical settings, with none of them being sufficient to cause problems |
| Understanding causal behaviors and functions | Connect information about system structures to behaviors and functions | "Physical Form: hollow bottoms; Physical Function: insufficient ground contact on conveyor belt; Generalized Function: dysfunctional transport of chocolate bars; Abstract Function: frequent faults at carrier belt; Functional Purpose: low efficiency" |
| Illusion of understanding | Provide information about actual system relations | "Temperature in the molding unit and time in the buffer both lead to warm chocolate, which can cause soiling. However, the two parameters can compensate each other" |
| | Prompt operators to think about the system in more specific terms | Show slow motion video to show how bars move back and forth between injector and transfer finger, instead of just informing operators that bars break in the injection unit |
| | Describe problems on different abstraction levels and allow switching between them | "Physical Form: hollow bottoms; Physical Function: insufficient ground contact on conveyor belt; Generalized Function: dysfunctional transport of chocolate bars; Abstract Function: frequent faults at carrier belt; Functional Purpose: low efficiency" |

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
