# Peer review of "Data in Context: How Digital Transformation Can Support Human Reasoning in Cyber-Physical Production Systems"

_futureinternet, doi:10.3390/fi13060156_

Round 1

Reviewer 1 Report

Interest work on context generation and use in production systems. 

Comments:

  • Minor spellcheck needed.
  • In the abstract the authors raise two points being addressed in this work: a) illustrate how the technologies can support contextual reasoning and b) highlight challenges. While the answers are included in the text, it would be highly suggested the authors summarizing them in the conclusion.
  • The literature review on CPPS is rather limited. It is suggested to extend it and maybe link it with recent studies on Semantic AI and/or linked data.
  • Please check and rephrase the first sentence in the introduction as its meaning is not clear. Also, regarding the following text "in the process and discrete processing..." consider rephrasing as follows "in the continuous and discrete processing ..."
  • Figures 1 and 2 seem blur. Please replace with ones of higher resolution
  • A context-aware interconnected production system, one can argue that it would be easier to control or automate. In conventional production paradigms, human knowledge and experience would benefit the production system through bottom-up innovation. However, the impact or benefit of contextual information and reasoning to a human worker/supervisor/other and from his/her point of view and not production's is not clear. Please include a short paragraph elaborating on this. 

Author Response

Interest work on context generation and use in production systems. 

Thank you very much for your positive evaluation and constructive feedback!

Comments:

  • Minor spellcheck needed.

One of our authors (David Humphrey) is a native English speaker and writes texts for a living. He has read and edited the text, and we feel that his English language skills are adequate to produce a readable text. However, if you or the journal employees generating the proofs have any specific suggestions for words or sentences that you feel we should change, we are happy to do so.

  • In the abstract the authors raise two points being addressed in this work: a) illustrate how the technologies can support contextual reasoning and b) highlight challenges. While the answers are included in the text, it would be highly suggested the authors summarizing them in the conclusion.

This is a very good point, and we have changed the conclusion accordingly.

  • The literature review on CPPS is rather limited. It is suggested to extend it and maybe link it with recent studies on Semantic AI and/or linked data.

We have added a section about CPPS in the Introduction (p.1, lines 68-89), which explains what CPPS are and mentions some of the relevant technologies. However, we have not gone into much detail there, because the presentation of the specific technologies is what our technology part (section 4) is all about.

  • Please check and rephrase the first sentence in the introduction as its meaning is not clear. Also, regarding the following text "in the process and discrete processing..." consider rephrasing as follows "in the continuous and discrete processing ..."

We have changed the first sentence to “In the process industries and discrete processing industries, operators’ activities of process monitoring and process control can be characterized as problem solving.” The version you suggested sounds good as well, but it would not be quite correct due to two points:

(1) There is a difference between “process industries” and “(discrete) processing industries” (see Müller & Oehm, 2019). While the first one refers to a transformation of mainly fluid raw materials into other substances in chemical processes, the second one refers to a transformation of mainly three-dimensional objects into more complex objects in mechanical processes (e.g., production of pizzas, packaging of cheese).

(2) In the process industries, two types can be distinguished: continuous and batch. Thus, referring to all of the process industries as “continuous” would be misleading.

  • Figures 1 and 2 seem blur. Please replace with ones of higher resolution

The figures themselves have a very high resolution and the blurring occurred when we inserted them into the text. However, we have uploaded the original image files to the journal website. Thus the journal staff can use it accordingly.

  • A context-aware interconnected production system, one can argue that it would be easier to control or automate. In conventional production paradigms, human knowledge and experience would benefit the production system through bottom-up innovation. However, the impact or benefit of contextual information and reasoning to a human worker/supervisor/other and from his/her point of view and not production's is not clear. Please include a short paragraph elaborating on this. 

This is a great point, thanks for bringing it up! We have added a paragraph about this issue in the Discussion (p. 20, lines 925-931).

Reviewer 2 Report

FOCUS: The authors review psychological literature in four areas relevant to contextualization: information sampling, integration, categorization, and causal reasoning.  Based on the literature, authors derive functional requirements  for digital transformation technologies, focusing on the cognitive activities they should support.

Strong Points: 

1. The paper addresses a very important aspect of research in cyber-physical systems.
1. Contextual data is important for systems e.g CPS that can help to improve a number of aspect. It can also be used to detect real-time failures in CPSs.

Potential Improvements:

1. The paper need a proof read.
2. As mentioned above, the contextual data can also be used to detect failures in CPSs. I recommend to add following papers in your related work which use contextual data to predict failures in CPSs. " Failure Detection and Prevention for Cyber-Physical Systems Using Ontology-Based Knowledge Base. Computers 2018, 7, 68. https://doi.org/10.3390/computers7040068"

Author Response

FOCUS: The authors review psychological literature in four areas relevant to contextualization: information sampling, integration, categorization, and causal reasoning.  Based on the literature, authors derive functional requirements  for digital transformation technologies, focusing on the cognitive activities they should support.

Strong Points: 

1. The paper addresses a very important aspect of research in cyber-physical systems.
1. Contextual data is important for systems e.g CPS that can help to improve a number of aspect. It can also be used to detect real-time failures in CPSs.

Thank you very much for taking the time to review our manuscript and helping us to improve it! We were happy to read your positive evaluation and were grateful for the constructive comments.

Potential Improvements:

1. The paper need a proof read.

One of our authors (David Humphrey) is a native English speaker and writes texts for a living. He has read and edited the text, and we feel that his English language skills are adequate to produce a readable text. However, if you or the journal employees generating the proofs have any specific suggestions for words or sentences that you feel we should change, we are happy to do so.

  1. As mentioned above, the contextual data can also be used to detect failures in CPSs. I recommend to add following papers in your related work which use contextual data to predict failures in CPSs. " Failure Detection and Prevention for Cyber-Physical Systems Using Ontology-Based Knowledge Base. Computers2018, 7, 68. https://doi.org/10.3390/computers7040068"

Thank you for the interesting reference! We have included it both in the Introduction (section 1.2) where we introduce CPPS and in the Discussion (section 5.2) where we argue that contextualization is not only useful for the operator but also for the technical system itself.

Reviewer 3 Report

Thank you for the opportunity to review this paper. While the subject certainly seems of interest to the field, it has a number of issues which I will try to discuss in more detail below:

[1] Most importantly: I am not sure, whether this review meets the criteria for systematic reviews. The research questions are quite broad and the quality of some included studies seems to be low; also they are not RCTs, hence, in a strict sense, the objective of classical systematic reviews answering the question of efficacy may not be answered by the current review (and is not intended to be answered, as I understand). Hence, a scoping review may be the more appropriate form.

[2] The presentation of results also does not seem to conform to common standards. It is redundant, and should only include the presentation of findings and not interpretations. The latter has to be reserved for the discussion section.

[3] Language needs to be checked by a native speaker as there are some unclear/imprecise expressions and grammatical errors throughout the manuscript. 

[4] Paragraphs in the paper are longer. Generally, a paragraph should be at least four sentences. The basic rule for determining paragraph length is to keep each paragraph to only one main idea. If a paragraph contains multiple ideas, it is likely that the ideas aren’t fully explained or supported (in other words, the paragraph isn’t fully developed). It is hard to read the sections of the paper.

[5] The introduction section is too large. This section Introduction establishes the scope, context, and significance of the research being conducted by summarizing current understanding about the topic, stating the purpose of the work in the form of the research problem supported by a hypothesis or a set of questions, explaining briefly the methodological approach used to examine the research problem, highlighting the potential outcomes your study can reveal, and outlining the remaining structure and organization of the paper.  

[6] The review should follow the proposed PRISMA guidelines for reviews to assure comparability and transparency: http://www.prisma-statement.org/. Explain the criteria and methodology for exclusion of the papers found at the beginning of the systematic review. 

[7] The scope and content of the review are defined quite broadly and contain a cursory overview over the subject (rather than assessing its effectiveness or efficacy), also, the quality/type of included studies do not sufficiently allow to answer the question of effectiveness. . In fact, in order to emphasize the novelty of the provided content, the most relevant reviews on the topic should be cited and the differential contributions of the provided article (i.e., the contributions that cannot be found in other papers) should be highlighted.

[6] Consider including also other databases in your search, i.e. SCOPUS, PUBMED, or google scholar. Also, was grey literature searched? (see opengrey.eu)

[7] Include more references to support and strengthen your statements/conclusions/predictions in your discussion section

[8] The study poses vaguely formulated questions. The authors do not elaborate in the introduction what is the purpose of their review in terms of (i) other similar reviews, (ii) future research directions, and (iii) identification of research gaps.

[9] The inclusion of papers in the study does not take into account their relevance (for instance, by means of their impact factor or number of citations).

[10] The conclusions of the work are qualitative responses without new contributions and relevance to the state of the art.

[11] ] The references are inadequate because some of them are too old and there not follow the Citations Style Guide for example 1, 6, 7, 9, 10, 12, 13, 14, 15, and so on. The articles cited must be within the past five years

This work is not considered with enough quality, novelty, or relevance to recommend its publication. This type of study should focus on relevant, measurable contributions supported by the data obtained in the review

Author Response

Thank you for the opportunity to review this paper. While the subject certainly seems of interest to the field, it has a number of issues which I will try to discuss in more detail below:

Thank you for taking so much time and effort to review the paper. We really appreciate this. However, it seems like many of your critical comments arise from a miscommunication about the purpose of the paper. We will explain this in more detail in response to the specific points you brought up.

[1] Most importantly: I am not sure, whether this review meets the criteria for systematic reviews. The research questions are quite broad and the quality of some included studies seems to be low; also they are not RCTs, hence, in a strict sense, the objective of classical systematic reviews answering the question of efficacy may not be answered by the current review (and is not intended to be answered, as I understand). Hence, a scoping review may be the more appropriate form.

This paper certainly does not meet the criteria for a systematic review, because it is not intended to be a systematic review. Actually, it is not even a scoping review, because as far as we understand it, the purpose of a scoping review still is to synthesize evidence about a particular research topic.

In contrast, our aim is to highlight a number of psychological issues that should be considered when designing human-technology cooperation in cyber-physical production systems, and a number of technologies that could be helpful in addressing them. For each of these issues (45 in total, e.g., sampling biases) you could write a systematic review on its own, whereas in our paper each of them is merely introduced in a few sentences. We certainly do not summarize all the research that is relevant with respect to one issue (e.g., all evidence about sampling biases) but merely state that the issue exists, and briefly explain what it is. Thus, our paper is a necessarily subjective collection of issues that can be considered in depth in later work with regard to its impacts on human-machine interaction and specific technological measures to address it.

One can argue that this type of paper should not exist, because only systematic reviews about individual (psychological or technological) issues are valuable. However, in total we present 45 psychological issues. We do not think that an engineer can or should be expected to read dozens of psychological reviews before joining an interdisciplinary discussion about human-machine cooperation in CPPS. Therefore, we felt that in order to provide a suitable starting point for an interdisciplinary discussion, we needed this broad collection of issues and their relevance with regard to the application of digital transformation technologies.

In the revised version of the article, we have made this more explicit. Moreover, to avoid confusion we have changed all parts of the text that used the word “review”, and replaced it by phrases such a “presenting examples”.

[2] The presentation of results also does not seem to conform to common standards. It is redundant, and should only include the presentation of findings and not interpretations. The latter has to be reserved for the discussion section.

A consequence of the fact that we do not collect and synthesize evidence in this paper is that we cannot apply the common standards for a systematic review. Neither section 2 (cognitive challenges) nor section 4 (technologies) represents a collection of results that we can simply present and then integrate in the discussion. This is because the technologies in section 4 are possible answers to the challenges raised in section 2. They cannot be selected in a meaningful way without first interpreting the psychological issues and deriving requirements from them. Therefore, we have decided not to change the structure of our article.

[3] Language needs to be checked by a native speaker as there are some unclear/imprecise expressions and grammatical errors throughout the manuscript. 

One of our authors (David Humphrey) is a native English speaker and writes texts for a living. He has read and edited the text, and we feel that his English language skills are adequate to produce a readable text. However, if you or the journal employees generating the proofs have any specific suggestions for words or sentences that you feel we should change, we are happy to do so.

[4] Paragraphs in the paper are longer. Generally, a paragraph should be at least four sentences. The basic rule for determining paragraph length is to keep each paragraph to only one main idea. If a paragraph contains multiple ideas, it is likely that the ideas aren’t fully explained or supported (in other words, the paragraph isn’t fully developed). It is hard to read the sections of the paper.

We agree with you that paragraphs are an important issue. It is interesting to see that there are several contradictory rules of thumb to determine paragraph length. The one I learned was that there should be about three paragraphs per page. However, we think that paragraph length should mainly be determined by content, not by any content-free rule. Therefore, when writing the paper we have paid particular attention to that, ensuring that one paragraph corresponds to one content unit. We do not want to split up paragraphs just to make them shorter while distorting their meaning. Accordingly, we went through the whole text again to see where we could split paragraphs to address your concern, but it turned out that this only was possible in very few places.

[5] The introduction section is too large. This section Introduction establishes the scope, context, and significance of the research being conducted by summarizing current understanding about the topic, stating the purpose of the work in the form of the research problem supported by a hypothesis or a set of questions, explaining briefly the methodological approach used to examine the research problem, highlighting the potential outcomes your study can reveal, and outlining the remaining structure and organization of the paper.  

We have added subheadings to the Introduction to make it easier to keep an overview. However, we do not want to shorten the Introduction. This probably also has to do with the nature of our paper (i.e., not being a systematic review of one particular topic, for which it would indeed be easy to write a brief introduction).

You correctly note that in a systematic review, the Introduction establishes the scope, context and significance “by summarizing current understanding about the topic”. However, we do not have an individual topic for which we can then simply collect all relevant research. We aim to connect work from different disciplines in a way that might be helpful in solving current problems in CPPS. Therefore, our introduction must serve the following functions: It needs to characterize the current problem in production plants, it needs to introduce the innovation of CPPS, and it needs to explain the scope of the present article and distinguish it from other research.

Moreover, you mention that the Introduction should be supported by a hypothesis. Again, this makes perfect sense for a systematic review. However, we are wondering how you think this could work given the scope of our paper, being a collection of examples of cognitive challenges, an extraction of requirements, and a matching with relevant example technologies. We are curious what you think could be a hypothesis for the collection of psychological challenges, and how it would be possible to test it.

It seems to us that your points, while being perfectly valid for a particular type of article (i.e., systematic review), cannot be applied to the present type of work. Therefore, we have not made any major changes to the Introduction.

[6] The review should follow the proposed PRISMA guidelines for reviews to assure comparability and transparency: http://www.prisma-statement.org/. Explain the criteria and methodology for exclusion of the papers found at the beginning of the systematic review. 

These guidelines apply to systematic reviews of a particular research area and thus are not applicable to the present work. For instance, our selection of articles about each individual cognitive issue is not only incomplete. Rather, it is mostly just one or two article which are used to highlight that the issue exists. Accordingly, we cannot specify criteria for including or excluding articles, because they are only examples, anyway. How would it make sense to specify the exclusion criteria for articles when you only include one or two (of dozens or even hundreds) to represent each issue?

[7] The scope and content of the review are defined quite broadly and contain a cursory overview over the subject (rather than assessing its effectiveness or efficacy), also, the quality/type of included studies do not sufficiently allow to answer the question of effectiveness. . In fact, in order to emphasize the novelty of the provided content, the most relevant reviews on the topic should be cited and the differential contributions of the provided article (i.e., the contributions that cannot be found in other papers) should be highlighted.

Indeed, the scope of the article is quite broad, and would certainly be too broad if it was a review of one particular research topic. Similar to your earlier points, we think that the criteria of effectiveness and efficacy might apply to systematic reviews, but not to a collection of example challenges and potential technologies used to address these challenges.

Concerning the aims of the present work, we think that we have been quite explicit, and there is an entire page dedicated to explaining the aims and scope. At the start of this section, we state the following question: How can digital transformation technologies support the context-dependent selection and integration of data from different sources and thereby support categorization and causal reasoning?

This question includes (a) the four research areas from which we extract cognitive challenges, as well as (b) the question how these issues can be supported by technologies. What else would you need in terms of specificity?

Concerning other related papers, there is a paragraph that cites some of them and provides a detailed explanation of three ways in which our article differs from them. We do not want to dedicate more space to that, because whereas these papers address operator support in CPPS, they do not focus on the psychological challenges of contextualization (or even reasoning in a broader sense), and thus are not particularly relevant to our work.

[6] Consider including also other databases in your search, i.e. SCOPUS, PUBMED, or google scholar. Also, was grey literature searched? (see opengrey.eu)

Our paper is not a systematic review and thus is not based on a comprehensive search process. Given that we cited only one or two exemplar articles as a reference for our collection of the 45 psychological issues, we do not see where the search of different databases or the inclusion of grey literature would play a role in that. Would it be to prove that the one or two articles we selected are the most important ones for the respective issue? We do not think that it is possible to determine that.

[7] Include more references to support and strengthen your statements/conclusions/predictions in your discussion section

We cannot quite understand this criticism, because in the original version of the article, 46 references were cited in the Discussion (in the revised version it is a little more than 50). We think this is an acceptable number. Therefore, we have decided not to include more references. If you think a specific reference should be included to strengthen a specific statement, we are happy to add it.

[8] The study poses vaguely formulated questions. The authors do not elaborate in the introduction what is the purpose of their review in terms of (i) other similar reviews, (ii) future research directions, and (iii) identification of research gaps.

Concerning your points about the research question and related literature, please see our replies to comment [7]. Research gaps and future research questions are addressed in section 4.3 (Limitations and future work). In this section, we have now added a subsection about specific implementations – that is, choosing a particular psychological issue and asking how a specific technology could be used to address it, or vice versa.

Moreover, we want to emphasize that the purpose of this article is not to identify research gaps in a particular research area. Instead, in our view the very broad research gap is that the psychological and technological issues have rarely been brought together in the past. This can hardly be considered a gap, because the things that are missing by far outweigh the bits and fragments that might exist in some isolated applications. Thus, instead of highlighting research gaps, our message is that psychologists and engineers should start working together on the design of contextualization support for CPPS – actually creating a new research area instead of filling gaps. 

[9] The inclusion of papers in the study does not take into account their relevance (for instance, by means of their impact factor or number of citations).

We believe that the relevance of an article is a matter of its content and its fit to the subject matter, not its impact factor and citations. In fact, we consider the inclusion based on citation or impact metrics to be in conflict with standards of good scientific practice. Here is an article that discusses responsible ways of using references, and selecting them by non-content criteria such as citation count, impact factor or age is not one of them: https://journals.plos.org/ploscompbiol/article?id=10.1371/journal.pcbi.1006036

Penders B (2018) Ten simple rules for responsible referencing. PLoS Comput Biol 14(4): e1006036. https://doi.org/10.1371/journal.pcbi.1006036

Therefore, we have not and certainly will not use such criteria in deciding which literature should be cited in the paper.

[10] The conclusions of the work are qualitative responses without new contributions and relevance to the state of the art.

We regret that you see it like this, but we wonder what else, other than qualitative responses, you would expect in relation to which state of the art. Given that the article is not intended to be a systematic review of any psychological or technological issue, our aim is not to contribute to the state of the art in any of the individual issues.

With regard to contextualization support in CPPS, we are surprised that you do not find a contribution to the state of the art in the paper. After all, we have generated a list of 108 support strategies and outlined a large number of technologies that are relevant in implementing them. We did not find comparable information anywhere in the literature, and thus consider it to be a meaningful contribution. However, if you know any state of the art about contextualization support for operators of CPPS, we would be more than happy if you could let us know about this work.

[11] ] The references are inadequate because some of them are too old and there not follow the Citations Style Guide for example 1, 6, 7, 9, 10, 12, 13, 14, 15, and so on. The articles cited must be within the past five years

Thank you for your comment about the citation style guide – we have changed the reference list accordingly.

Concerning the age of the references, we have used new references where they were suitable. However, we completely disagree with the statement that the articles that are cited must be within the past five years. If relevant research has been conducted earlier, we do not understand why this research should be ignored, and other less relevant research should be cited instead. Basically, this seems to imply that the entire body of scientific evidence would have to be recollected every five years. We can understand that a focus on the latest research is important when the goal is to provide an overview of the state of the art in a technological area. However, especially in psychology we have lots of older articles that still are highly relevant today.

This work is not considered with enough quality, novelty, or relevance to recommend its publication. This type of study should focus on relevant, measurable contributions supported by the data obtained in the review

In fact, measurable contributions supported by data are not what this review is intended to contribute. We hope that we were able to explain the intended contribution somewhat more clearly now. Moreover, we are curious whether after a consideration of our aims and their contrast to the aims of a systematic review, you will still consider our research irrelevant. If yes, we would be delighted to hear your ideas about other ways that make it possible to achieve our goal (i.e., building bridges between disciplines by starting from a problem and presenting a broad overview of the relevant concepts from each domain, linked by psychological requirements). Moreover, we are curious about the state of the art you mentioned, because we were unable to find any state of the art concerning the human-centered use of technologies for the contextualization of information in CPPS.

Round 2

Reviewer 1 Report

No further comments

Reviewer 3 Report

The authors addressed all my concerns in this revised version. I appreciate it a lot for their effort.